# SURF: Separation via Unsupervised Remixing Flow

**Henry Li** [* ⋆ 1]  **Robin Scheibler** [* 2]  **Efthymios Tzinis** [1]  **Matt Shannon** [2]  **Arnaud Doucet** [† 2]  **John R. Hershey** [† 2]

## Abstract

The goal of single-channel source separation is to reconstruct $K$ sources given their mixture. In supervised settings where vast amounts of clean source data are available, this challenging, ill-posed problem has been addressed successfully by generative diffusion and flow-based prior models. However, access to such clean source samples is often limited, and even when available, supervised models are vulnerable to domain shifts. To bridge this gap, we present Separation via Unsupervised Remixing Flow (**SURF**), an unsupervised flow matching approach for source separation that learns directly from observed mixtures. This method relies on a novel combination of state-of-the-art supervised flow matching and regression-based self-supervised techniques. At a high level, starting from a teacher model, we utilize a "remixing" step to bootstrap the learning of a student flow model from the teacher's estimates. We provide insights into the objectives optimized by this approach and draw a novel connection to the Wake-Sleep algorithm. Empirical evaluations on image and audio benchmarks demonstrate that **SURF** establishes a new state-of-the-art, significantly outperforming existing unsupervised methods. See our demo page for examples.

## 1. Introduction

Single-channel source separation is a fundamental challenge in signal processing, where one must recover a set of underlying signals given a single mixture. As this problem is highly ill-posed, deep learning approaches have historically relied on discriminative training, where models such as Conv-TasNet (Luo & Mesgarani, 2019) or Transformers (Wang et al., 2023; Saijo et al., 2024) are trained to map

---

[*]Equal contribution [†]Equal senior contribution [⋆]Work partly done as a student researcher at Google DeepMind [1]Google [2]Google DeepMind. Correspondence to: Henry Li <lihenry@google.com>.

*Proceedings of the $43^{rd}$ International Conference on Machine Learning*, Seoul, South Korea. PMLR 306, 2026. Copyright 2026 by the author(s).

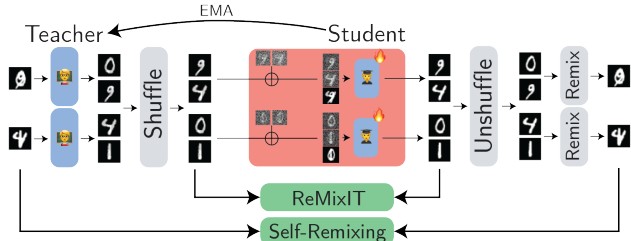

*Figure 1.* Illustration of **SURF**. Given initial mixtures, a teacher model first produces source estimates. These are shuffled, then used as self-supervised examples to a student flow matching model. The student is trained to predict the estimated sources (ReMixIT) or original mixtures (Self-Remixing).

mixtures directly to sources. While these regression-based techniques have been remarkably successful, they suffer from two critical limitations: they often introduce unrealistic artifacts (Larsen et al., 2016), and they rely on vast amounts of clean labeled data.

To specifically mitigate regression artifacts and better capture the distribution of natural signals, the field has pivoted towards generative models (Subakan & Smaragdis, 2018; Jayaram & Thickstun, 2020). In particular, diffusion and flow-based models (Ho et al., 2020; Song et al., 2021; Lipman et al., 2023) have enabled the learning of high-quality priors for clean sources. Common approaches in this domain treat separation as a conditional generation task, utilizing either mixture-conditional diffusion in the mel-spectral domain (Chen et al., 2023) or multi-source time-domain models with guidance terms for mixture consistency (Mariani et al., 2024; Janati et al., 2025; Shi et al., 2026). Diffusion has also been effectively applied to post-process and refine estimates from conventional regression models (Wang et al., 2024). Other methods employ non-standard noising dynamics (Scheibler et al., 2023; Dong et al., 2025) or flow matching (Scheibler et al., 2025; Shi et al., 2025).

Despite these developments, a critical bottleneck remains. All the aforementioned approaches operate under the assumption that clean source data is accessible, either directly or through a pre-trained diffusion prior. It is worth noting that while several methods adopt the "unsupervised" terminology, they do rely on such a pre-trained prior, a dependency our work explicitly avoids. In many complex domains such as bio-acoustics (Xie et al., 2021), hyperspectral imaging (Bioucas-Dias et al., 2012), or gravitational

wave detection (Christensen, 2018), obtaining such clean, in-domain source data is practically impossible. Moreover, even in scenarios where clean source data can be obtained for prior training, domain shift between the training set and real-world measurements remains a pervasive challenge.

To address the lack of clean source data, unsupervised regression-based source separation techniques have been developed; in particular, **MixIT** has become prominent (Wisdom et al., 2020). To further enhance performance of MixIT, regression-based self-supervised approaches such as **ReMixIT** (Tzinis et al., 2022) and **Self-Remixing** (Saijo & Ogawa, 2023; Saijo et al., 2025) have been proposed. These methods operate on a "teacher-student" principle: a teacher model estimates sources from a mixture, and these estimates are then permuted and summed to form synthetic mixtures with known pseudo-targets; i.e., the teacher source estimates. The student model is then trained to separate these synthetic mixtures, effectively bootstrapping its performance.

Earlier attempts to learn variational autoencoders priors from noisy data for source separation (Neri et al., 2021) had limited success. More recently, proposals to learn diffusion-based priors directly from corrupted data using Expectation-Maximization have emerged (Rozet et al., 2024; Hosseintabar et al., 2025). These methods require approximating complex guidance terms or training a conditional diffusion model at each iteration to learn an unconditional model for clean sources, which can be computationally prohibitive. Furthermore, these unsupervised generative approaches have not yet been applied to single-channel source separation. The method we present here bypasses having to train an unconditional model.

In this work, we propose Separation via Unsupervised Remixing Flow (**SURF**), a novel framework that integrates state-of-the-art supervised flow matching (Scheibler et al., 2025) with regression-based self-supervised remixing (Tzinis et al., 2022; Saijo & Ogawa, 2023). This integration is non-trivial: while remixing algorithms rely on combining deterministic source estimates to enforce consistency, flow matching operates by learning velocity fields. SURF bridges this structural discrepancy, enabling the training of generative techniques directly from mixtures; see Figure 1 for a high-level illustration.

We provide an analysis that sheds new light on the objectives optimized by existing self-supervised methods (Tzinis et al., 2022; Saijo & Ogawa, 2023; Saijo et al., 2025). We further show that an instance of **SURF** can be reinterpreted as a Wake-Sleep algorithm (Hinton et al., 1995), a classical method for training latent-variable generative models. Finally, we demonstrate the efficacy of our unsupervised generative approach on image and audio separation benchmarks, showing state-of-the-art results.

## 2. Problem Formulation & Background

### 2.1. Source Separation

Source separation can be understood as an inverse problem where given the observed mixture

$$\boldsymbol{m} = \sum_{k=1}^{K} \boldsymbol{x}^{(k)}, \tag{1}$$

we are tasked with recovering the $K$ underlying signals $\boldsymbol{x}^{(k)} \in \mathbb{R}^{1 \times d}$. We write $\boldsymbol{x} = [\boldsymbol{x}^{(1)\top} \cdots \boldsymbol{x}^{(K)\top}]^{\top} \in \mathbb{R}^{K \times d}$.

Inverting (1) is non-trivial due to two primary factors.

**Single-channel Measurements** When $\boldsymbol{m}$ is a $K$-channel measurement and $\boldsymbol{x}^{(k)}$ are noise-free, this formulation becomes the independent component analysis problem with well-known results (Jutten & Herault, 1991; Comon, 1994; Hyvärinen & Oja, 2000). However, in our case $\boldsymbol{m}$ and $\boldsymbol{x}^{(k)}$ are assumed to be *single-channel*. In this setting, recovering the sources from this single mixture observation becomes a highly ill-posed problem, necessitating a strong prior on the underlying signals.

**Permutation Invariance** Henceforth, we will assume that the individual sources are i.i.d., i.e.,

$$p(\boldsymbol{x}) = p(\boldsymbol{x}^{(1)}, \dots, \boldsymbol{x}^{(K)}) = \prod_{k=1}^{K} p(\boldsymbol{x}^{(k)}). \tag{2}$$

This implies that $p(\boldsymbol{x})$ is invariant to the class of all $K$-permutations denoted $S_K$. In other words, there is no single canonical ordering of the individual sources $\{\boldsymbol{x}^{(k)}\}_{k=1}^{K}$. As the likelihood $p(\boldsymbol{m}|\boldsymbol{x}) = \delta(\boldsymbol{m} - \sum_{k=1}^{K} \boldsymbol{x}^{(k)})$ induced by (1) is also permutation invariant, this implies that the posterior $p(\boldsymbol{x}|\boldsymbol{m}) \propto p(\boldsymbol{m}|\boldsymbol{x})p(\boldsymbol{x})$ is also permutation invariant.

To treat this invariance, a standard approach in source separation is to use a permutation invariant training (PIT) wrapper (Isik et al., 2016; Kolbæk et al., 2017). Given an estimate $\hat{\boldsymbol{x}}$ of $\boldsymbol{x}$, PIT modifies any loss function $\ell$ via

$$\mathcal{L}(\boldsymbol{x}, \hat{\boldsymbol{x}}) = \min_{\sigma \in S_K} \ell(\hat{\boldsymbol{x}}, \sigma \boldsymbol{x}). \tag{3}$$

where $\sigma$ is identified as a matrix permuting the rows of $\boldsymbol{x}$.

### 2.2. Supervised Source Separation via Flow Matching

In the supervised case, we have access to clean source samples $\boldsymbol{x} \sim p(\boldsymbol{x})$ (see (2)) and thus to $\boldsymbol{m} = \sum_{k=1}^{K} \boldsymbol{x}^{(k)}$; i.e., samples from $p(\boldsymbol{x}, \boldsymbol{m})$. We review here how conditional Flow Matching (FM) can be applied to solve source separation in this setting (Lipman et al., 2023) as proposed by Scheibler et al. (2025). FM defines an Ordinary Differential Equation (ODE) $(\boldsymbol{x}_t)_{t \in [0,1]}$ which, initialized at $t = 0$ from a reference "noise" distribution

$x_0 \sim p_0(x_0|m)$, outputs at time $t = 1$ samples $x_1 \sim p_1(x_1|m) = p(x_1|m) \propto p(x_1, m)$. To achieve this, we define a probability path $(p_t(\cdot|m))_{t \in [0,1]}$ that interpolates smoothly between $p_0(\cdot|m)$ and $p_1(\cdot|m)$: $p_t(\cdot|m)$ being given by the marginal distribution of $x_t = (1-t)x_0 + tx_1$ for $x_0 \sim p_0(\cdot|m)$, $x_1 \sim p_1(\cdot|m)$. For source separation, the noise distribution for $x_0 \in \mathbb{R}^{K \times d}$ is defined for $z \in \mathbb{R}^{K \times d}$ by

$$x_0 = \frac{1}{K}\mathbb{1}m + P^\perp z, \text{ for } \text{vec}(z) \sim \mathcal{N}(0, I_{Kd}), \quad (4)$$

where $P^\perp = I_K - P$ for $P = \mathbb{1}\mathbb{1}^\top/K$ and $\mathbb{1}$ is the all one column vector of size $K$ so that $P^\perp$ acts on the source index on the left. This ensures that $m = \mathbb{1}^\top x_0$ and yields $x_1 - x_0 = P^\perp(x_1 - z)$.

The drift of the flow matching ODE is then obtained by minimizing the following loss

$$\mathcal{L}_{\text{FM}}(\theta) = \mathbb{E}\big[\|v_\theta(x_t, t, m) - (x_1 - x_0)\|^2\big], \quad (5)$$

where the expectation is w.r.t. $t \sim \mathcal{U}(0,1)$, $m \sim p(m)$, $x_1 \sim p_1(\cdot|m)$, (or, equivalently, $x_1 \sim p(\cdot)$ and $m = \mathbb{1}^\top x_1$), $x_0 \sim p_0(\cdot|m)$ and $x_t = (1-t)x_0 + tx_1$. For an expressive drift family, the solution of this minimization problem is $v_{\theta^*}(x, t, m) = \mathbb{E}[x_1 - x_0 \mid x_t = x, m] := u(x, t, m)$ and is such that for $x_0 \sim p_0(\cdot|m)$ and

$$dx_t = u(x_t, t, m)dt \quad (6)$$

then $x_t \sim p_t(\cdot|m)$ for $t \in [0,1]$, in particular $x_1 \sim p_1(\cdot|m)$; see Lipman et al. (2023).

To exploit the mixture consistent and permutation invariant properties of the source separation task arising from (1) and (2), a projection layer is used to ensure mixture consistent outputs for any $v_\theta$, i.e., $m = \mathbb{1}^\top x_1$, and we rely on a permutation equivariant architecture (Scheibler et al., 2025).

To further improve performance, Scheibler et al. (2025) proposed the following alternative inspired by PIT (see (3)). Given $x_0, x_1$, one selects

$$\sigma_{x_0}^{x_1}(\theta) = \arg\min_{\sigma \in S_K} \|v_\theta(x_0, 0, m) - (\sigma x_1 - x_0)\|^2 \quad (7)$$

$$= \arg\min_{\sigma \in S_K} \|\hat{x}_{1,\theta}(x_0, m) - \sigma x_1\|^2, \quad (8)$$

for the estimate of the clean sources

$$\hat{x}_{1,\theta}(x_0, m) = x_0 + v_\theta(x_0, 0, m). \quad (9)$$

We abuse notation and write $\sigma := \sigma_{x_0}^{x_1}(\theta)$ but we underline that $\sigma$ is a function of $x_0, x_1$ and $\theta$. Scheibler et al. (2025) then considers the modified flow matching loss

$$\mathcal{L}_{\text{MFM}}(\theta) = \mathbb{E}\big[\|v_\theta(x_t^\sigma, t, m) - (\sigma x_1 - x_0)\|^2\big] \quad (10)$$

for the interpolant

$$x_t^\sigma = (1-t)x_0 + t\sigma x_1, \quad (11)$$

$\sigma$ being identified as a matrix permuting the rows of $x_1$ and a stop-gradient being used on $\sigma$ in (10). Despite $\sigma$ being a function of $x_0, x_1$ and $\theta$, it can be shown that $\sigma x_1$ is marginally distributed according to $p_1(\cdot|m)$ for any $m$ when $x_0 \sim p_0(\cdot|m)$, $x_1 \sim p_1(\cdot|m)$; see Appendix C.1 for details. This implies that the interpolant (11) defines a valid flow matching path between $p_0(\cdot|m)$ and $p_1(\cdot|m)$. The resulting method, **FLOSS**, outperforms regular flow matching and achieves state-of-the-art in supervised separation (Scheibler et al., 2025). It will be instrumental in the development of the proposed techniques.

### 2.3. Unsupervised Source Separation

ReMixIT (Tzinis et al., 2022) and Self-Remixing (Saijo & Ogawa, 2023) improve upon the prominent unsupervised MixIT method (Wisdom et al., 2020). These self-supervised methods train a student model using new mixtures created by recombining sources extracted by a teacher; see Figure 1.

Let $M = [m_1^\top \cdots m_B^\top]^\top$ be a batch of $B$ mixtures, $f_\mathcal{T}$ be a teacher model and $\bar{x}_b = f_\mathcal{T}(m_b)$ the estimate of the sources in the $b^{\text{th}}$ mixture. These sources are recombined into new mixtures with a permutation matrix $\Pi$ sampled uniformly on the symmetric group $S_{BK}$

$$\widetilde{X} = \Pi\bar{X}, \text{ where } \begin{cases} \widetilde{X} = [\tilde{x}_1^\top \cdots \tilde{x}_B^\top]^\top, \\ \bar{X} = [\bar{x}_1^\top \cdots \bar{x}_B^\top]^\top, \end{cases} \quad (12)$$

so $\bar{X}, \widetilde{X}$ are of dimension $BK \times d$. We then create a batch of $B$ new mixtures of dimension $B \times d$ using

$$\widetilde{M} = (I_B \otimes \mathbb{1}^\top) \, \widetilde{X}, \quad (13)$$

where $\otimes$ is the Kronecker product.

Now let $\hat{x}_b = f_\theta^b(\tilde{m})$ be the estimate of the sources in the $b^{\text{th}}$ new mixture by a student model. In ReMixIT (Tzinis et al., 2022), $f_\theta$ is trained using the loss

$$\mathcal{L}_{\text{RM}}(\theta) = \mathbb{E}\left[\left\|\hat{X} - \Upsilon\,\widetilde{X}\right\|^2\right] = \mathbb{E}\left[\left\|\Upsilon^{-1}\,\hat{X} - \widetilde{X}\right\|^2\right] \quad (14)$$

where $\Upsilon$ is chosen to attribute each source estimate to its original source using a PIT loss (3); i.e. $\Upsilon = \text{blkdiag}(\sigma_1, \ldots, \sigma_B)$ where $\sigma_b = \arg\min_{\sigma \in S_K} \|\hat{x}_b - \sigma\tilde{x}_b\|^2$. One issue with this loss is that the targets $\widetilde{X}$ include the errors from the teacher model. Self-Remixing bypasses this issue by regressing w.r.t. *the input mixtures*,

$$\mathcal{L}_{\text{SR}}(\theta) = \mathbb{E}\left[\left\|(I_B \otimes \mathbb{1}^\top)\Pi^{-1}\Upsilon^{-1}\,\hat{X} - M\right\|^2\right]. \quad (15)$$

After having trained the student, we update the teacher parameters using an exponential moving average mechanism before updating again the student.

Both ReMixIT (Tzinis et al., 2022) and Self-Remixing (Saijo & Ogawa, 2023) provide a framework for teacher models $f_{\mathcal{T}}$ to be distilled into student models. This process can then be repeated by using the student model as the new teacher model. This results empirically in a student model that exceeds the performance of the original teacher model.

## 3. Separation via Unsupervised Remixing Flow

We propose Separation via Unsupervised Remixing Flow (**SURF**), a novel *generative* unsupervised separation algorithm via Flow Matching (FM). **SURF** integrates the supervised flow matching **FLOSS** method of (Scheibler et al., 2025) with regression-based self-supervised methods (Tzinis et al., 2022; Saijo & Ogawa, 2023; Saijo et al., 2025). We will use **FLOSS** for both the teacher and the student, initialized by a MixIT model (Wisdom et al., 2020). To achieve this, we will rely on the key connection between denoiser and drift for flow matching.

### 3.1. Relating Flow Matching Velocity to Regression

A significant discrepancy between FM and regression-based approaches is that flow models estimate a velocity term $\boldsymbol{v}_\theta$ at $\boldsymbol{x}_t$, whereas regression-based models directly estimate the source $\hat{\boldsymbol{x}}$ itself. To bridge these concepts, we use the following elementary result

$$\mathbb{E}[\boldsymbol{x}_1 \mid \boldsymbol{x}_t, \boldsymbol{m}] = \boldsymbol{x}_t + (1-t)\boldsymbol{u}(\boldsymbol{x}_t, t, \boldsymbol{m}) \quad (16)$$
$$\approx \boldsymbol{x}_t + (1-t)\boldsymbol{v}_\theta(\boldsymbol{x}_t, t, \boldsymbol{m}). \quad (17)$$

which follows from $\boldsymbol{u}(\boldsymbol{x}, t) = \mathbb{E}[\boldsymbol{x}_1 - \boldsymbol{x}_0 | \boldsymbol{x}_t = \boldsymbol{x}, \boldsymbol{m}]$ (Lipman et al., 2023) and $\boldsymbol{x}_1 - \boldsymbol{x}_0 = \frac{\boldsymbol{x}_1 - \boldsymbol{x}_t}{1-t}$.

### 3.2. ReMixIT for Flow Matching

ReMixIT for FM can be thought of as applying supervised FM to synthetic mixtures created by randomly mixing the source estimates given by a teacher model. Specifically, given teacher outputs $\overline{\boldsymbol{X}}$ obtained by FM based on mixture samples $\boldsymbol{M}$, we sample a random permutation $\boldsymbol{\Pi}$ and shuffle the $BK$ rows of $\overline{\boldsymbol{X}}$ to obtain $\widetilde{\boldsymbol{X}}_1 = \widetilde{\boldsymbol{X}} = \boldsymbol{\Pi}\overline{\boldsymbol{X}}$ as in (12). These shuffled sources are used to create synthetic mixtures using $\widetilde{\boldsymbol{M}} = (\boldsymbol{I}_B \otimes \mathbb{1}^\top) \ \widetilde{\boldsymbol{X}}_1$. We then apply supervised flow matching to $\widetilde{\boldsymbol{X}}_1$. Following (4), we sample initial conditions $\widetilde{\boldsymbol{X}}_0 = \frac{1}{K}(\boldsymbol{I}_B \otimes \mathbb{1}) \ \widetilde{\boldsymbol{M}} + (\boldsymbol{I}_B \otimes \boldsymbol{P}^\perp)\boldsymbol{Z}$ where $\boldsymbol{Z} \in \mathbb{R}^{BK \times d}$, $\mathrm{vec}(\boldsymbol{Z}) \sim \mathcal{N}(0, \boldsymbol{I}_{BKd})$.

Let $\boldsymbol{\Upsilon} = \mathrm{blkdiag}(\sigma_1, \ldots, \sigma_B)$ where $\sigma_b \in S_K$. We define the batch version of an interpolant of the form (11)

$$\widetilde{\boldsymbol{X}}_t^{\boldsymbol{\Upsilon}} = (1-t) \ \widetilde{\boldsymbol{X}}_0 + t\boldsymbol{\Upsilon} \ \widetilde{\boldsymbol{X}}_1 \quad (18)$$

The set of $B$ permutations $\boldsymbol{\Upsilon}$ we use is obtained by minimizing a PIT-type loss as in (7)

$$\boldsymbol{\Upsilon} = \operatorname*{arg\,min}_{\boldsymbol{\Gamma} \in (S_K)^B} \|\boldsymbol{v}_\theta(\widetilde{\boldsymbol{X}}_0, 0, \widetilde{\boldsymbol{M}}) - (\boldsymbol{\Gamma} \ \widetilde{\boldsymbol{X}}_1 - \widetilde{\boldsymbol{X}}_0)\|^2 \quad (19)$$

for $\boldsymbol{\Gamma} = \mathrm{blkdiag}(\gamma_1, \ldots, \gamma_B)$, i.e.

$$\sigma_b = \operatorname*{arg\,min}_{\gamma_b \in S_K} ||\hat{\boldsymbol{x}}_{1,\theta}(\tilde{\boldsymbol{x}}_{0,b}, \tilde{\boldsymbol{m}}_b) - \gamma_b \tilde{\boldsymbol{x}}_{1,b}||^2. \quad (20)$$

In (19), we write the stacked velocities of dimension $BK \times d$ for any $\boldsymbol{\Upsilon} \in (S_K)^B$ as the vertical block stack

$$\boldsymbol{v}_\theta(\widetilde{\boldsymbol{X}}_t^{\boldsymbol{\Upsilon}}, t, \widetilde{\boldsymbol{M}}) := \begin{bmatrix} \boldsymbol{v}_\theta(\tilde{\boldsymbol{x}}_{t,1}^{\sigma_1}, t, \tilde{\boldsymbol{m}}_1) \\ \vdots \\ \boldsymbol{v}_\theta(\tilde{\boldsymbol{x}}_{t,B}^{\sigma_B}, t, \tilde{\boldsymbol{m}}_B) \end{bmatrix} \in \mathbb{R}^{BK \times d}.$$

We will denote

$$\boldsymbol{R}_t := \boldsymbol{v}_\theta(\widetilde{\boldsymbol{X}}_t^{\boldsymbol{\Upsilon}}, t, \widetilde{\boldsymbol{M}}) - (\boldsymbol{\Upsilon} \ \widetilde{\boldsymbol{X}}_1 - \widetilde{\boldsymbol{X}}_0). \quad (21)$$

So the equivalent of the FM loss (10) applied to data $\widetilde{\boldsymbol{X}}$ we will minimize is simply given by

$$\mathcal{L}_{\text{RM-FM}}(\theta) = \mathbb{E}\left[\|\boldsymbol{R}_t\|^2\right], \quad (22)$$

where $\|\cdot\|$ is used to denote (abusively) the Frobenius norm. The expectation is w.r.t. $t \sim \lambda$ (a normalized weighting function), $\boldsymbol{M}, \boldsymbol{\Pi}, \widetilde{\boldsymbol{X}}_0, \widetilde{\boldsymbol{X}}_1$. The corresponding algorithm is presented in Algorithm 1 (red).

### 3.3. Self-Remixing for Flow Matching

Now consider the Self-Remixing loss, we want to adapt (15) to our flow matching framework. Leveraging (17), we propose the clean source estimates

$$\widehat{\boldsymbol{X}}_{1,\theta}^{\boldsymbol{\Upsilon}}(\widetilde{\boldsymbol{X}}_t^{\boldsymbol{\Upsilon}}, t, \widetilde{\boldsymbol{M}}) = \widetilde{\boldsymbol{X}}_t^{\boldsymbol{\Upsilon}} + (1-t)\boldsymbol{v}_\theta(\widetilde{\boldsymbol{X}}_t^{\boldsymbol{\Upsilon}}, t, \widetilde{\boldsymbol{M}}), \quad (23)$$

where $\widetilde{\boldsymbol{X}}_t^{\boldsymbol{\Upsilon}}$ is defined by (18) and (19).

Plugging $\widehat{\boldsymbol{X}}_{1,\theta} := \widehat{\boldsymbol{X}}_{1,\theta}^{\boldsymbol{\Upsilon}}(\widetilde{\boldsymbol{X}}_t^{\boldsymbol{\Upsilon}}, t, \widetilde{\boldsymbol{M}})$ into (15) gives us the self-remixing loss by taking the expectation over all the random variables and $t \sim \mathcal{U}[0,1]$. This loss can be re-expressed in a form more similar to (22) by writing

$$\left\|(\boldsymbol{I}_B \otimes \mathbb{1}^\top)\boldsymbol{\Pi}^{-1}\boldsymbol{\Upsilon}^{-1} \widehat{\boldsymbol{X}}_{1,\theta} - \boldsymbol{M}\right\|^2 \quad (24)$$

$$= \left\|(\boldsymbol{I}_B \otimes \mathbb{1}^\top)\boldsymbol{\Pi}^{-1}\big(\underbrace{\boldsymbol{\Upsilon}^{-1} \widehat{\boldsymbol{X}}_{1,\theta}}_{\text{Student}} - \underbrace{\widetilde{\boldsymbol{X}}_1}_{\text{Target}}\big)\right\|^2, \quad (25)$$

where the mixture consistency of the teacher yields $\boldsymbol{M} = (\boldsymbol{I}_B \otimes \mathbb{1}^\top)\boldsymbol{\Pi}^{-1} \widetilde{\boldsymbol{X}}_1$.

Now, using (18), (21) and (23), we obtain

$$\boldsymbol{\Upsilon}^{-1} \widehat{\boldsymbol{X}}_{1,\theta} - \widetilde{\boldsymbol{X}}_1 = (1-t)\boldsymbol{\Upsilon}^{-1}\boldsymbol{R}_t. \quad (26)$$

**Algorithm 1** SURF (ReMixIT and Self-Remixing)

1: **Require:** Flow Matching Model, Teacher $\theta_{\mathcal{T}}$, Student $\theta$, Batch $B$, EMA parameter $\alpha$
2: **while** not converged **do**
3:     $\boldsymbol{M} \sim p(\boldsymbol{M})$                **Unsupervised Remixing**
4:     $\bar{\boldsymbol{X}} \sim p_{\theta_{\mathcal{T}}}(\bar{\boldsymbol{X}} \mid \boldsymbol{M})$
5:     $\boldsymbol{\Pi} \sim \mathcal{U}(S_{BK}), \quad \widetilde{\boldsymbol{X}}_1 = \boldsymbol{\Pi}\bar{\boldsymbol{X}}$
6:     $\widetilde{\boldsymbol{M}} = (\boldsymbol{I}_B \otimes \mathbb{1}^{\top}) \, \widetilde{\boldsymbol{X}}_1$

---

7:     $t \sim \lambda$                **Flow Matching Path & Loss**
8:     $\text{vec}(\boldsymbol{Z}) \sim \mathcal{N}(\boldsymbol{0}, \boldsymbol{I}_{BKd})$ for $\boldsymbol{Z} \in \mathbb{R}^{BK \times d}$
9:     $\widetilde{\boldsymbol{X}}_0 = \frac{1}{K}(\boldsymbol{I}_B \otimes \mathbb{1}) \, \widetilde{\boldsymbol{M}} + (\boldsymbol{I}_B \otimes \boldsymbol{P}^{\perp})\boldsymbol{Z}$
10:    $\boldsymbol{\Upsilon} = \arg\min_{\boldsymbol{\Gamma}} \|\boldsymbol{v}_{\theta}(\widetilde{\boldsymbol{X}}_0, 0, \widetilde{\boldsymbol{M}}) - (\boldsymbol{\Gamma} \, \widetilde{\boldsymbol{X}}_1 - \widetilde{\boldsymbol{X}}_0)\|^2$
11:    $\widetilde{\boldsymbol{X}}_t^{\boldsymbol{\Upsilon}} = (1 - t) \, \widetilde{\boldsymbol{X}}_0 + t\boldsymbol{\Upsilon} \, \widetilde{\boldsymbol{X}}_1$
12:    $\boldsymbol{R}_t := \boldsymbol{v}_{\theta}(\widetilde{\boldsymbol{X}}_t^{\boldsymbol{\Upsilon}}, t, \widetilde{\boldsymbol{M}}) - (\boldsymbol{\Upsilon} \, \widetilde{\boldsymbol{X}}_1 - \widetilde{\boldsymbol{X}}_0)$
13:    $\mathcal{L}_{\text{RM-FM}} = \|\boldsymbol{R}_t\|^2$
14:    $\mathcal{L}_{\text{SR-FM}} = \|(\boldsymbol{I}_B \otimes \mathbb{1}^{\top})\boldsymbol{\Pi}^{-1}\boldsymbol{\Upsilon}^{-1}\boldsymbol{R}_t\|^2$

---

15:    $\theta \leftarrow \theta - \eta\nabla_{\theta}\mathcal{L}_{\text{RM-FM}/\text{SR-FM}}$     **Update**
16:    $\theta_{\mathcal{T}} \leftarrow \alpha\theta_{\mathcal{T}} + (1 - \alpha)\theta$
17: **end while**

Combining (25) and (26) first gives the regression-style loss induced by the denoising proxy (23):

$$\mathcal{L}_{\text{SR-FM}}^{\text{aux}}(\theta) = \mathbb{E}\left[(1 - t)^2 \left\|(\boldsymbol{I}_B \otimes \mathbb{1}^{\top})\boldsymbol{\Pi}^{-1}\boldsymbol{\Upsilon}^{-1}\boldsymbol{R}_t\right\|^2\right]. \tag{27}$$

In the algorithm we instead use the following reweighted objective

$$\mathcal{L}_{\text{SR-FM}}(\theta) = \mathbb{E}\left[\left\|(\boldsymbol{I}_B \otimes \mathbb{1}^{\top})\boldsymbol{\Pi}^{-1}\boldsymbol{\Upsilon}^{-1}\boldsymbol{R}_t\right\|^2\right]. \tag{28}$$

This is equivalent to the objective induced by the endpoint proxy (95) discussed in Appendix C.3. Recall that (24) shows that the regression target of this loss depends on the teacher estimates $\widetilde{\boldsymbol{X}}_1$ only through $\boldsymbol{M}$.

The corresponding algorithm is presented in Algorithm 1 (green).

## 4. Analysis

Here, we provide insights into the objectives optimized by these algorithms. Existing results for standard ReMixIT are limited (see Tzinis et al., 2022) and, to the best of our knowledge, non-existent for Self-Remixing. Because our FM framework introduces additional complexity, we introduce simplifying assumptions to make the analysis tractable.

### 4.1. Alignment and Remixing in the Population Limit

We consider a population analysis version where the batch size $B \to \infty$. Let $p(\boldsymbol{m})$ denote the true mixture distribution. Since source tuples are unordered, we use the following alignment convention only for the population analysis. Given the true source tuple $\boldsymbol{x} = [\boldsymbol{x}^{(1)\top} \cdots \boldsymbol{x}^{(K)\top}]^{\top}$ and a "raw" teacher sample $\bar{\boldsymbol{x}}^{\text{raw}} \sim p_{\theta_{\mathcal{T}}}(\cdot \mid \boldsymbol{m})$, choose a PIT minimizer

$$\pi_{\mathcal{T}}(\boldsymbol{x}, \bar{\boldsymbol{x}}^{\text{raw}}) \in \arg\min_{\pi \in S_K} \sum_{k=1}^{K} \left\|\bar{\boldsymbol{x}}^{\text{raw},(\pi(k))} - \boldsymbol{x}^{(k)}\right\|^2,$$

and define $\bar{\boldsymbol{x}}^{(k)} = \bar{\boldsymbol{x}}^{\text{raw},(\pi_{\mathcal{T}}(k))}$. Thus $\bar{\boldsymbol{x}}^{(k)}$ denotes the teacher estimate aligned with the unobserved true source $\boldsymbol{x}^{(k)}$. This is a relabelling convention used only to define population error terms, not an algorithmic step.

For finite $B$, the batch permutation $\boldsymbol{\Pi} \in S_{BK}$ is used to approximate independent recombination of teacher sources drawn from different mixtures. In the $B \to \infty$ limit, this is captured by the *population* marginal of a single teacher-estimated source:

$$\bar{p}_{\theta_{\mathcal{T}}}(\bar{\boldsymbol{x}}^{(k)}) = \int p_{\theta_{\mathcal{T}}}(\bar{\boldsymbol{x}}^{(k)}, \bar{\boldsymbol{x}}^{(-k)} \mid \boldsymbol{m})p(\boldsymbol{m})\mathrm{d}\bar{\boldsymbol{x}}^{(-k)}\mathrm{d}\boldsymbol{m}, \tag{29}$$

where $(-k)$ denotes all sources but the $k$th. This distribution is independent of $k$ by exchangeability. For each teacher estimate $\bar{\boldsymbol{x}}^{(k)}$ for $k \in \{1, \ldots, K\}$, sample "background" sources $\bar{\boldsymbol{x}}_{\text{bg}}^{(k,2)}, \ldots, \bar{\boldsymbol{x}}_{\text{bg}}^{(k,K)} \overset{\text{i.i.d.}}{\sim} \bar{p}_{\theta_{\mathcal{T}}}$. We then form the matrix "anchored" to source $k$

$$\tilde{\boldsymbol{x}}_k := \left[(\bar{\boldsymbol{x}}^{(k)})^{\top}, (\bar{\boldsymbol{x}}_{\text{bg}}^{(k,2)})^{\top}, \ldots, (\bar{\boldsymbol{x}}_{\text{bg}}^{(k,K)})^{\top}\right]^{\top} \in \mathbb{R}^{K \times d}, \tag{30}$$

and write $\tilde{\boldsymbol{m}}_k := \mathbb{1}^{\top}\tilde{\boldsymbol{x}}_k$ for the corresponding synthetic mixture. This construction is the population replacement of the batch shuffle $\boldsymbol{\Pi}$ and the regrouping $(\boldsymbol{I}_B \otimes \mathbb{1}^{\top}) \, \widetilde{\boldsymbol{X}}_1$: no explicit $\boldsymbol{\Pi}$ appears, only i.i.d. sampling from $\bar{p}_{\theta_{\mathcal{T}}}$.

We then also sample $\tilde{\boldsymbol{x}}_{k,0} \sim p_0(\boldsymbol{x}_0 \mid \tilde{\boldsymbol{m}}_k)$ following (4). We consider the clean source estimates as in (9)

$$\hat{\boldsymbol{x}}_{k,1,\theta}(\tilde{\boldsymbol{x}}_{k,0}, \tilde{\boldsymbol{m}}_k) = \tilde{\boldsymbol{x}}_{k,0} + \boldsymbol{v}_{\theta}(\tilde{\boldsymbol{x}}_{k,0}, 0, \tilde{\boldsymbol{m}}_k). \tag{31}$$

For each matrix $\tilde{\boldsymbol{x}}_k$, define the PIT permutation as in (7)

$$\sigma_k = \arg\min_{\sigma \in S_K} \|\hat{\boldsymbol{x}}_{k,1,\theta}(\tilde{\boldsymbol{x}}_{k,0}, \tilde{\boldsymbol{m}}_k) - \sigma\tilde{\boldsymbol{x}}_k\|^2. \tag{32}$$

### 4.2. Population-Level Unsupervised Flow Matching

Let us denote

$$\boldsymbol{r}_k = \boldsymbol{v}_{\theta}(\tilde{\boldsymbol{x}}_{k,t}^{\sigma_k}, t, \tilde{\boldsymbol{m}}_k) - (\sigma_k\tilde{\boldsymbol{x}}_k - \tilde{\boldsymbol{x}}_{k,0}). \tag{33}$$

The population analogue of the ReMixIT-Flow objective (22) is given by

$$\mathcal{L}_{\text{RM-FM}}^{\infty}(\theta) = \frac{1}{K}\mathbb{E}\left[\|\boldsymbol{r}_k\|^2\right], \tag{34}$$

where $\tilde{x}_{k,t}^{\sigma} = (1-t)\tilde{x}_{k,0} + t\sigma\tilde{x}_k$ and the expectation is w.r.t. $\boldsymbol{m}$, $(\boldsymbol{x}^{(k)})_{k=1}^{K}$, $((\bar{\boldsymbol{x}}_{\text{bg}}^{(k,j)})_{j=2}^{K})_{k=1}^{K}$, $\tilde{\boldsymbol{x}}_{k,0}$, and $t \sim \lambda$. The left hand side of (34) is identical for all $k$.

Self-Remixing differs from ReMixIT in that supervision is against the observed mixture $\boldsymbol{m}$, and routing back to $\boldsymbol{m}$ is achieved by construction in (30): for source $k$, the "anchor" source $\bar{\boldsymbol{x}}^{(k)}$ occupies row 1 of $\tilde{\boldsymbol{x}}_k$ (the remaining rows are independent background sources).

For each matrix $\tilde{\boldsymbol{x}}_k$, we also define the PIT population (32). The predicted contribution to the original mixture is then the first row of $\tilde{\boldsymbol{x}}_k$, which is estimated by the first row of $\sigma_k^{-1}\hat{\boldsymbol{x}}_{k,1,\theta}^{\sigma_k}(\tilde{\boldsymbol{x}}_{k,t}^{\sigma_k}, t, \tilde{\boldsymbol{m}}_k)$. Here, the time-dependent estimate of the clean sources is defined by

$$\hat{\boldsymbol{x}}_{k,1,\theta}^{\sigma_k}(\tilde{\boldsymbol{x}}_{k,t}^{\sigma_k}, t, \tilde{\boldsymbol{m}}_k) = \tilde{\boldsymbol{x}}_{k,t}^{\sigma_k} + (1-t)\boldsymbol{v}_\theta(\tilde{\boldsymbol{x}}_{k,t}^{\sigma_k}, t, \tilde{\boldsymbol{m}}_k).$$

This is the population equivalent of (23). Hence, an estimate of the original mixture $\boldsymbol{m} = \sum_{k=1}^{K}\boldsymbol{x}^{(k)}$ is

$$\hat{\boldsymbol{m}}_\theta(\tilde{\boldsymbol{x}}_t, t, \tilde{\boldsymbol{m}}) = e_1^\top \sum_{k=1}^{K} \sigma_k^{-1}\hat{\boldsymbol{x}}_{k,1,\theta}^{\sigma_k}(\tilde{\boldsymbol{x}}_{k,t}^{\sigma_k}, t, \tilde{\boldsymbol{m}}_k),$$

where $e_1$ is the first standard basis vector. Giving teacher mixture consistency, $\sum_{k=1}^{K}\bar{\boldsymbol{x}}^{(k)} = \sum_{k=1}^{K}\boldsymbol{x}^{(k)} = \boldsymbol{m}$, calculations very similar to Section 3.3 yield

$$(1-t)^{-1}\left(\hat{\boldsymbol{m}}_\theta(\tilde{\boldsymbol{x}}_t, t, \tilde{\boldsymbol{m}}) - \boldsymbol{m}\right) = \sum_{k=1}^{K} e_1^\top \sigma_k^{-1}\boldsymbol{r}_k, \quad (35)$$

where $\boldsymbol{r}_k$ represents the error in the predicted drift for the specific anchored source $\bar{\boldsymbol{x}}^{(k)}$.

Using the same reweighted convention as in (28), and normalizing by $1/K$, we consider for $t \sim \lambda$

$$\mathcal{L}_{\text{SR}-\text{FM}}^{\infty}(\theta) = \frac{1}{K}\mathbb{E}\left[\left\|\sum_{k=1}^{K} e_1^\top \sigma_k^{-1}\boldsymbol{r}_k\right\|^2\right]. \quad (36)$$

### 4.3. Relationship to Supervised Flow Matching

We now use the alignment convention introduced above. Teacher estimates have already been relabelled with respect to the latent source tuple, so quantities such as $e^{(k)} = \bar{\boldsymbol{x}}^{(k)} - \boldsymbol{x}^{(k)}$ are well defined. For each remixed pseudo-example, the student output is also an unordered $K$-tuple; we route it by the PIT permutation associated with the matrix $\tilde{\boldsymbol{x}}_k$. After applying this routing, we suppress the explicit permutation notation. Thus, when we write $\hat{\boldsymbol{x}}_{\theta,t}^{(k)}$, we mean the student estimate routed to the anchor $\bar{\boldsymbol{x}}^{(k)}$, not a fixed raw output index.

We propose in Appendix C.2 an analysis of the regression-style ReMixIT and Self-Remixing and show how this analysis carries over to flow matching once we work with the

time-dependent clean estimate $\hat{\boldsymbol{x}}_{k,1,\theta}(\tilde{\boldsymbol{x}}_{k,t}, t, \tilde{\boldsymbol{m}}_k) = \tilde{\boldsymbol{x}}_{k,t} + (1-t)\boldsymbol{v}_\theta(\tilde{\boldsymbol{x}}_{k,t}, t, \tilde{\boldsymbol{m}}_k)$ and the anchor component $\hat{\boldsymbol{x}}_{\theta,t}^{(k)} := e_1^\top \hat{\boldsymbol{x}}_{k,1,\theta}(\tilde{\boldsymbol{x}}_{k,t}, t, \tilde{\boldsymbol{m}}_k)$. Let $e^{(k)} = \bar{\boldsymbol{x}}^{(k)} - \boldsymbol{x}^{(k)}$ and define the anchor sigma-field $\mathcal{A}_k = \sigma(\boldsymbol{x}^{(k)}, \bar{\boldsymbol{x}}^{(k)}, t)$. The systematic anchor-component error is $\delta_{\theta,t}^{(k)} := \mathbb{E}\big[\hat{\boldsymbol{x}}_{\theta,t}^{(k)} - \boldsymbol{x}^{(k)} \mid \mathcal{A}_k\big]$, i.e. the error averaged over the random background sources and the FM noise. Denoting for $\beta_t = (1-t)^{-2}\lambda_t$ the pseudo-supervised flow matching loss

$$L_{\text{Sup-FM}}^{\infty}(\theta) = \mathbb{E}\left[\beta_t \left\|\hat{\boldsymbol{x}}_{\theta,t}^{(1)} - \boldsymbol{x}^{(1)}\right\|^2\right]. \quad (37)$$

We call this the pseudo-supervised FM loss as $\hat{\boldsymbol{x}}_{\theta,t}^{(k)}$ is a function of $\tilde{\boldsymbol{m}}_k$ which does *not* follow the distribution of the observed mixtures.

We then get

$$L_{\text{RM-FM}}^{\infty}(\theta) = L_{\text{Sup-FM}}^{\infty}(\theta) + \mathbb{E}[\beta_t(\|e^{(1)}\|^2 - 2\langle e^{(1)}, \delta_{\theta,t}^{(1)}\rangle)],$$

$$L_{\text{SR}-\text{FM}}^{\infty}(\theta) = L_{\text{Sup-FM}}^{\infty}(\theta) + (K-1)\mathbb{E}[\beta_t \langle \delta_{\theta,t}^{(1)}, \delta_{\theta,t}^{(2)}\rangle],$$

where $t \sim \mathcal{U}[0,1]$. See Proposition C.4 for a formal statement and more complete results.

These decompositions suggest different mechanisms for ReMixIT and Self-Remixing. For ReMixIT, the correction term involves the teacher error $e^{(1)}$; hence, when the teacher is accurate or when its error is weakly aligned with the sensitivity of the background-averaged student error, this term should have limited influence on the gradient. For Self-Remixing, the correction term is of a different nature: it contains no explicit teacher-error factor. Its gradient is $2(K-1)\mathbb{E}\left[\beta_t(\nabla_\theta \delta_{\theta,t}^{(1)})^\top \delta_{\theta,t}^{(2)}\right]$ by exchangeability. Thus its size is controlled by the magnitude and cross-correlation of the systematic errors $\delta_{\theta,t}^{(k)}$ after averaging over the independently remixed background sources. This term is expected to be small when these background-averaged errors are small or weakly correlated across independently remixed anchors. In that regime, the dominant gradient contribution remains the pseudo-supervised FM term.

### 4.4. Wake-Sleep Interpretation of ReMixIT

We present here a probabilistic interpretation of ReMixIT as an instance of *Wake–Sleep* variational inference (Hinton et al., 1995) with an *implicit prior*. Wake–Sleep training is typically described for a latent variable model with a *prior* over latents and an *inference* (posterior) model.

Specialized to source separation, consider the generative model

$$\bar{p}_{\theta_T}(\bar{\boldsymbol{x}}, \boldsymbol{m}) = \bar{p}_{\theta_T}(\bar{\boldsymbol{x}})\, p(\boldsymbol{m} \mid \bar{\boldsymbol{x}}), \quad (38)$$

$$\bar{p}_{\theta_T}(\boldsymbol{m}) = \int \bar{p}_{\theta_T}(\bar{\boldsymbol{x}})\, p(\boldsymbol{m} \mid \bar{\boldsymbol{x}})\, d\bar{\boldsymbol{x}}, \quad (39)$$

*Table 1.* Comparison of methods on MNIST and CIFAR10 datasets, evaluated over 5,000 overlapping mixtures of images (10,000 separated images) formed from each test set. The best unsupervised method is **bolded** and second best is underlined.

| Method | Properties | | MNIST (2 Source Mixtures) | | | | CIFAR10 (2 Source Mixtures) | | | |
| --- | --- | --- | --- | --- | --- | --- | --- | --- | --- | --- |
| | Unsupervised | Generative | PSNR ↑ | LPIPS ↓ | SSIM ↑ | FID ↓ | PSNR ↑ | LPIPS ↓ | SSIM ↑ | FID ↓ |
| Supervised Regression | × | × | 26.02 | 0.007 | 0.963 | 25.44 | 19.34 | 0.059 | 0.726 | 22.18 |
| Supervised Flow | × | ✓ | 37.44 | 0.001 | 0.992 | 19.47 | 20.38 | 0.032 | 0.777 | 9.601 |
| BASIS (Jayaram & Thickstun, 2020) | × | ✓ | 29.67 | 0.005 | 0.919 | 38.62 | 13.37 | 0.119 | 0.429 | 26.66 |
| MixIT | ✓ | × | 21.90 | 0.011 | 0.929 | 30.89 | 16.77 | 0.069 | 0.707 | 29.51 |
| Regression (ReMixIT) | ✓ | × | 22.81 | 0.011 | 0.915 | 30.55 | 17.40 | 0.078 | 0.647 | 28.44 |
| Regression (Self-Remixing) | ✓ | × | 23.13 | 0.010 | 0.910 | 28.14 | 17.51 | 0.082 | 0.655 | 29.12 |
| SURF (ReMixIT) | ✓ | ✓ | **37.26** | **0.001** | **0.992** | 19.57 | **19.73** | **0.036** | **0.756** | 14.83 |
| SURF (Self-Remixing) | ✓ | ✓ | 37.03 | **0.001** | 0.991 | **19.56** | 19.49 | 0.037 | 0.751 | **14.77** |

*Table 2.* Inception / FID Score of 25,000 separations (50,000 separated images) of two overlapping mixtures of CIFAR-10 images. Best is **bolded** and second best is underlined.

| Algorithm | Inception Score ↑ | FID ↓ |
| --- | --- | --- |
| Average | $7.18 \pm 0.08$ | 28.02 |
| BASIS | **$8.29 \pm 0.16$** | 22.12 |
| MixIT | $4.56 \pm 0.05$ | 31.27 |
| SURF (ReMixIT) | $8.20 \pm 0.08$ | 12.98 |
| SURF (Self-Remixing) | $8.25 \pm 0.07$ | **12.50** |

where $\bar{p}_{\theta_\mathcal{T}}(\bar{x}) = \prod_{k=1}^{K} \bar{p}_{\theta_\mathcal{T}}(\bar{x}^{(k)})$ is the implicit prior defined in (29) and $p(m \mid \bar{x}) = \delta(m - \mathbb{1}^\top \bar{x})$. We also consider an "inference" model $p_\theta(\bar{x} \mid m)$, our student model. Because $p(m \mid \bar{x})$ is singular, the KL expressions below are finite only when the inference model is also mixture consistent.

A natural objective is to fit the generative model marginal $\bar{p}_{\theta_\mathcal{T}}(m)$ to the data distribution $p(m)$. Wake–Sleep alternates two KL objectives to achieve this:

**Wake:** $\arg\min_{\theta_\mathcal{T}} \mathrm{KL}[p(m)\, p_\theta(\bar{x} \mid m) \,\|\, \bar{p}_{\theta_\mathcal{T}}(\bar{x})\, p(m \mid \bar{x})]$,

**Sleep:** $\arg\min_{\theta} \mathrm{KL}[\bar{p}_{\theta_\mathcal{T}}(\bar{x})\, p(m \mid \bar{x}) \,\|\, p(m)\, p_\theta(\bar{x} \mid m)]$.

**Wake phase.** The Wake loss to update $\theta_\mathcal{T}$ is equivalent to minimize

$$\mathrm{KL}\left[p(m)\, p_\theta(\bar{x} \mid m) \,\|\, \bar{p}_{\theta_\mathcal{T}}(\bar{x})\, p(m \mid \bar{x})\right] \quad (40)$$
$$= \mathbb{E}_{(\bar{x},m) \sim p(m)\, p_\theta(\bar{x}|m)}\left[-\log \bar{p}_{\theta_\mathcal{T}}(\bar{x})\right] + \mathrm{const}.$$

However, $\bar{p}_{\theta_\mathcal{T}}(\bar{x})$ is not available in closed form as it is the *aggregate posterior* of the inference model over data mixtures $p(m)$ at parameter $\theta_\mathcal{T}$. So it is difficult to implement such a Wake phase. Nevertheless, we know that if the model was sufficiently expressive then, at optimality, we would get

$$\bar{p}_{\theta_\mathcal{T}}(\bar{x}^{(k)}) = \int p_\theta(\bar{x}^{(k)}, \bar{x}^{(-k)}|m)p(m)\mathrm{d}\bar{x}^{(-k)}\mathrm{d}m.$$

Given (29), this gives the heuristic update of moving $\theta_\mathcal{T}$

toward $\theta$; it should not be interpreted as an exact closed-form minimizer of the Wake KL, since the aggregate prior is implicit. However, such update can be practically unstable. Instead, we use an EMA update so that $\theta$ can move quickly but $\theta_\mathcal{T}$ tracks it slowly.

**Sleep phase.** Optimizing $\theta$ reduces to maximum likelihood training of the inference model on synthetic pairs $(\bar{x}, m)$ sampled from the generative model:

$$\mathrm{KL}\left[\bar{p}_{\theta_\mathcal{T}}(\bar{x})\, p(m|\bar{x}) \,\|\, p(m)\, p_\theta(\bar{x}|m)\right] \quad (41)$$
$$= \mathbb{E}_{(\bar{x},m) \sim \bar{p}_{\theta_\mathcal{T}}(\bar{x})p(m|\bar{x})}\left[-\log p_\theta(\bar{x} \mid m)\right] + \mathrm{const}.$$

Thus the Sleep phase trains the separator/inference model on synthetic mixtures generated from the prior $\bar{p}_{\theta_\mathcal{T}}(\bar{x}, m)$. To sample from $\bar{p}_{\theta_\mathcal{T}}(\bar{x})$, it follows from (29) that we need to sample $K$ mixture samples from $p(m)$ and apply the teacher to each of them. We then define $\bar{x}^{(k)}$ as the $k^{\text{th}}$ teacher source estimate associated to the $k^{\text{th}}$ mixture and then we sum them to obtain a synthetic mixture sample $m$. This is *identical* to ReMixIT in the population case; see Section 4.1.

If $p_\theta(\bar{x} \mid m)$ admitted tractable likelihoods, the sleep update would minimize the cross-entropy in (41). In our case, $p_\theta$ is induced by FM and we minimize a standard FM surrogate. If we had a model sufficiently expressive, then at optimality we would get $p_\theta(\bar{x}|m) = \bar{p}_{\theta_\mathcal{T}}(\bar{x}|m)$. Note that this does not imply $\theta = \theta_\mathcal{T}$ in the general case.

## 5. Empirical Results

We evaluate here **SURF** (Algorithm 1) with both ReMixIT and Self-Remixing on four datasets across image (MNIST (LeCun & Cortes, 2010), CIFAR10 (Krizhevsky, 2009)) and audio domains (Libr2iMix (Cosentino et al., 2020) and AudioSet (Gemmeke et al., 2017)). Libr2iMix consists of single-source data samples that are formed into synthetic mixtures while AudioSet consists of 2M true audio mixtures from YouTube videos, with no ground truth available.

The teacher model is initially trained with MixIT (Wisdom

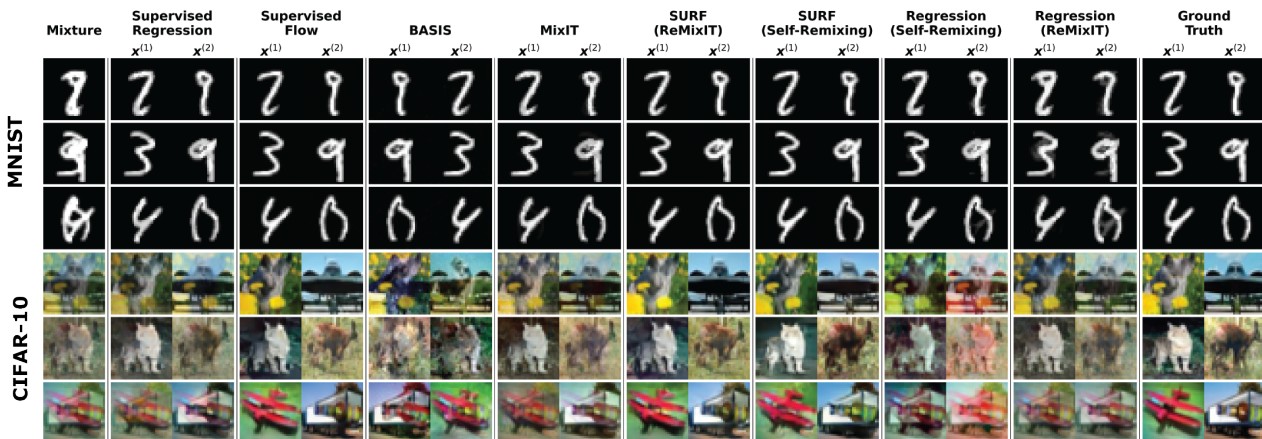

*Figure 2.* Qualitative examples for image separation on the MNIST (above) CIFAR10 (below) datasets, comparing Supervised Regression, Supervised Flow, BASIS, MixIT, and SURF algorithms. More results in Appendix E.

*Table 3.* Evaluation of methods trained on Audioset (Gemmeke et al., 2017). We evaluate on noisy speech samples (LibriSpeech + FUSS) and general multi-source separation (FUSS). The best unsupervised method is **bolded** and second best is underlined. The FUSS metrics are based on the SI-SDR for single (1S) or multiple (2Si, 3Si, 4Si) sources. See our demo page for listening examples.

| Method | FUSS (1 - 4 Source Mixtures) | | | | | | | LibriSpeech + FUSS (2 Source Mixtures) | | | |
|---|---|---|---|---|---|---|---|---|---|---|---|
| | 1S ↑ | 2Si ↑ | 3Si ↑ | 4Si ↑ | Under ↓ | Equal ↑ | Over ↓ | SI-SDR ↑ | ESTOI ↑ | PESQ ↑ | DNSMOS ↑ |
| Supervised Flow | 38.79 | 13.58 | 12.46 | 10.24 | 0.031 | 0.580 | 0.389 | 18.21 | 0.904 | 3.29 | 2.58 |
| MixIT | 10.99 | 9.20 | **11.87** | **9.75** | **0.005** | 0.339 | 0.656 | 14.18 | 0.771 | 2.79 | 2.53 |
| ReMixIT | 19.83 | 9.91 | 8.47 | 8.62 | 0.014 | 0.387 | 0.599 | 14.29 | 0.793 | 2.85 | 2.51 |
| Self-Remixing | 19.75 | 9.88 | 8.58 | 9.11 | 0.035 | 0.244 | 0.721 | 14.81 | 0.784 | 2.81 | 2.56 |
| SURF (ReMixIT) | **32.67** | 11.04 | 10.38 | 7.52 | 0.166 | **0.574** | **0.260** | 14.98 | **0.840** | 2.86 | 2.55 |
| SURF (Self-Remixing) | 29.10 | **11.36** | 11.63 | 8.80 | 0.129 | 0.559 | 0.312 | **15.23** | **0.840** | **2.93** | **2.57** |

*Table 4.* Evaluation of methods on the Libri2Mix evaluation dataset. The best unsupervised method is **bolded** and second best is underlined.

| Method | Libri2Mix (2 Source Mixtures) | | | |
|---|---|---|---|---|
| | SI-SDR ↑ | ESTOI ↑ | PESQ ↑ | DNSMOS ↑ |
| Supervised Flow | 17.89 | 0.908 | 3.45 | 2.94 |
| MixIT | 12.39 | 0.753 | 2.52 | 2.36 |
| ReMixIT | 13.22 | 0.827 | 2.72 | 2.76 |
| Self-Remixing | 13.60 | 0.830 | 2.71 | 2.78 |
| SURF (ReMixIT) | **16.54** | **0.893** | **3.30** | **2.94** |
| SURF (Self-Remixing) | 16.28 | 0.889 | 3.28 | 2.93 |

et al., 2020). The student model is then trained with a frozen teacher model. Finally, the teacher is dynamically updated by EMA with decay constant $\alpha < 1$; we additionally utilize a hybrid teacher sampler during this phase for training stability (see Appendix D). For all datasets, we compare against the initial teacher MixIT model, non-generative ReMixIT (Tzinis et al., 2022) and Self-Remixing (Saijo & Ogawa, 2023) baselines. Where supervised data is available, we also compare against supervised regression and flow matching separation models.

Quantitative evaluation is challenging because mixture de-

composition is often non-unique. While standard distance metrics like PSNR and SSIM (Wang et al., 2004) (images) or SI-SDR (Le Roux et al., 2019), PESQ (Rix et al., 2001), and ESTOI (Jensen & Taal, 2016) (audio) provide reconstruction-based fidelity measures, they are limited in capturing the quality of the separated samples. For a more holistic perspective, we follow Jayaram & Thickstun (2020) and Scheibler et al. (2025) and complement these with perceptual metrics: LPIPS (Zhang et al., 2018), Inception Score (IS), and Fréchet Inception Distance (FID) (Heusel et al., 2017) for images, and DNSMOS (Reddy et al., 2022) for audio. These metrics assess distributional resemblance, offering insight into generation quality beyond strict deviation from the ground truth. To evaluate universal (non-speech) source separation, we use the Free Universal Sound Separation (FUSS) metrics (Wisdom et al., 2021). They are based on the SI-SDR for one (1S) or multiple sources (MSi2–4). See Appendix D for more details about the audio metrics.

### 5.1. Image Separation

We evaluate image separation performance using the MNIST and CIFAR-10 datasets, following the experimental protocol established to evaluate BASIS (Jayaram & Thick-

stun, 2020). For both datasets, training and evaluation mixtures are constructed by averaging pairs of randomly selected images from the respective test sets. During MixIT training in the first phase, single source mixtures are also included, following (Wisdom et al., 2020). We evaluate on 5,000 overlapping mixtures, resulting in 10,000 separated images for analysis. All models are trained with the noise-conditional score network (NCSN) architecture, with additional training details provided in Appendix D.

As clean, single-source data is available, we also compare against supervised regression and flow matching separation models as reference benchmarks. The supervised regression models are trained with a log-SNR PIT loss (3) with $\ell(\boldsymbol{x}, \hat{\boldsymbol{x}}) = 10 \log_{10}(||\boldsymbol{x} - \hat{\boldsymbol{x}}||^2)$. The supervised flow matching model is trained via the ReMixIT variant of Algorithm 1, where the teacher model samples $\widetilde{\boldsymbol{X}}$ are replaced by ground truth sources. In Table 1, we find that both variants of **SURF** excel in separation quality, both in terms of distance-based metrics (PSNR and SSIM) and in terms of perceptual quality metrics (LPIPS and FID), and are only surpassed by the supervised FM baseline.

We further evaluate the learned model's approximation quality of an *aggregate posterior* as described in Section 4.4. We conduct a dedicated comparison to sample quality with Inception Score (IS) and FID, reference-free metrics of overall generated sample quality, on 50,000 images (25,000 image pairs). Table 2 shows that SURF performs comparably to BASIS, a supervised separation algorithm that leverages a diffusion prior trained purely on clean sources in terms of IS, and outperforms it in terms of FID.

### 5.2. Audio Separation

We conduct speech separation experiments on the Libri2Mix dataset, consisting of 10 second utterances from male and female speakers drawn from LibriSpeech recorded at 16kHz sample rate. We train on the *train-360-clean* split of Libri2Mix, which contains 364 hours of mixtures where source utterances are drawn without replacement, and evaluate on 3,000 mixtures from the test split.

In speech separation, supervised data is also available, so we compare against a supervised flow matching model. We find that SURF variants again compare favorably against regression-based unsupervised baselines across intrusive metrics (SI-SDR, ESTOI, PESQ) as well as perceptual metrics (DNSMOS), and approaches the separation quality of the supervised flow matching model (Table 4).

Next, we train a model for universal sound separation on AudioSet (Gemmeke et al., 2017), approximately 731 h of YouTube videos containing speech, music, and general human, animal, and environmental sounds. Ground truth isolated sources are not available for AudioSet.

We conduct two sets of evaluations on these models (Table 3). First, we gauge general sound separation quality on the FUSS dataset (Wisdom et al., 2021). We observe a contrast between the MixIT teacher and remixing based approaches. The former tends to over-separate, while the latter under-separate. We hypothesize this is due to MixIT being trained on mixtures-of-mixtures with double the average number of sources in the input. SURF largely outperforms the other methods on 1 and 2 sources, but does slightly worse on 3 and 4 sources. In addition, it significantly improves the Equal metric, suggesting the proposed method is better at estimating the number of sources in the mixture on average.

We then evaluate the speech enhancement capability of our universal model. We mix samples from the LibriSpeech (Panayotov et al., 2015) *test-clean* set with the *background* source in each FUSS mixture (Wisdom et al., 2021), at a randomly chosen SNR between (-5, 15) dB, then renormalizing to (-1, 1). Again, we find that SURF generalizes well to this task and compares favorably to existing approaches in quantitative metrics, further suggesting that SURF learns an implicit prior over sources solely from in-the-wild mixtures, enabling a variety of downstream tasks.

## 6. Conclusion

We introduced **SURF**, a novel generative approach that learns to separate sources directly from mixtures by combining Flow Matching with self-supervised remixing methods. We provided insights into the objectives optimized by this approach and proposed a probabilistic interpretation through the lens of the Wake-Sleep algorithm. Experiments on image and audio datasets demonstrate that **SURF** outperforms unsupervised baselines across most scenarios, delivering superior perceptual quality and reducing artifacts compared to regression models.

While complex mixtures with many sources remain a challenging problem, these findings establish **SURF** as an effective solution for source separation in real-world environments where clean training data is inaccessible.

## Impact Statement

The presented algorithm introduces a generative framework for large scale unsupervised separation. Source separation itself can be susceptible to adversarial exploitation (such as speech de-anonymization). Moreover, the generative approach proposed here requires careful consideration of the potential for hallucination, which may amplify data biases in downstream tasks and complicate the provenance of data processed by such a model. These issues are important to address in extensions of this work.

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

## A. Organization of the Supplementary Material

In Appendix B, we summarize the notation in the paper. In Appendix C, we first provide results establishing the validity of the supervised flow matching algorithm proposed in (Scheibler et al., 2025) in Appendix C.1. We then establish results for ReMixIT and Self-Remixing in C.2, first in the regression case in Appendix C.2.1 and then extend those results to the generative case in Appendix C.2.2. Finally, we establish an alternative derivation of the Self-Remixing loss for flow matching in Appendix C.3. Further experimental details are given in Appendix D. Finally, more experimental results can be found in Appendix E.

## B. Notation

We summarize the notation used in this text in Table 5. Our notation distinguishes between the global batch level view and the local mixture level view: bold uppercase variables (e.g., $M, X$) represent the full batch of dimension $BK \times d$ (or $B \times d$ for mixtures), while bold lowercase variables (e.g., $m_b, x_b$) refer to a specific $K$-source block or a single mixture at batch index $b$. In the population analysis (Section 4) where $B \to \infty$, we replace the batch index with the subscript $k$ to denote variables anchored to a specific source type (e.g., $\tilde{x}_k, \tilde{m}_k$). Noisy variables (e.g., $X_t$ or $x_{b,t}$) are denoted by an additional index $t$. Finally, a parenthetical superscript refers to a single source (e.g. $x_{t,b}^{(i)}$ is the $i$-th source of $x_{t,b}$).

We use diacritics to denote the stage of the SURF pipeline: $\overline{X}$ represents estimates generated by the fixed Teacher, $\widetilde{X}$ represents the randomly permuted "remixed" sources used as synthetic targets, and $\widehat{X}$ represents the predictions of the student FM model. The time state $t \in [0, 1]$ of the flow matching process is denoted in the subscript (e.g., $\widehat{X}_t$). Finally, we explicitly distinguish between the random remixing permutation $\Pi$ and the optimized alignment permutations ($\sigma_b, \Upsilon$) used for remixing and PIT-based alignment during flow model training (Section 3.2), respectively.

## C. Proofs

### C.1. Validity of Variants of Supervised Flow Matching

In (Scheibler et al., 2025), a variation of the flow matching loss is proposed but its validity has not been established rigorously. We close here this gap. They consider the following setup. Given $x_0 \sim p_0(\cdot|m), x_1 \sim p_1(\cdot|m)$, ones selects the $x_0, x_1, \theta$-dependent permutation

$$\sigma(x_0, x_1, \theta) = \arg\min_{\sigma \in S_K} ||v_\theta(x_0, 0, m) - (\sigma x_1 - x_0)||^2. \tag{42}$$

We then consider a flow matching loss

$$\mathcal{L}_{\text{MFM}}(\theta) = \mathbb{E}\Big[ \Big\| v_\theta(x_t^{\sigma(x_0, x_1, \theta)}, t, m) - (\sigma(x_0, x_1, \theta)x_1 - x_0) \Big\|^2 \Big] \tag{43}$$

where $x_t^\sigma := x_t^{\sigma(x_0, x_1, \theta)} = (1-t)x_0 + t\sigma(x_0, x_1, \theta)x_1$. A stop-gradient is used on $\sigma(x_0, x_1, \theta)$.

The following proposition shows that the joint distribution of $(x_0^\sigma, x_1^\sigma)$ is such that $x_0^\sigma = x_0 \sim p_0(\cdot|m)$ (which is obvious) but crucially, although the permutation depends on both the data and the model parameters, the symmetry assumptions imply that $x_1^\sigma \sim p_1(\cdot|m)$. Hence, this shows that $x_t^\sigma$ defines a valid flow matching path between $p_0(\cdot|m)$ and $p_1(\cdot|m)$.

For the sake of simplicity, we simplify notation and remove the conditioning by $m$.

**Proposition C.1.** *Let $x_0 \sim p_0$ and $x_1 \sim p_1$ be independent random variables taking values in $\mathcal{X} := (\mathbb{R}^d)^K$, equipped with the norm $\|x\|^2 := \sum_{k=1}^K \|x^{(k)}\|_{\mathbb{R}^d}^2$. The symmetric group $S_K$ acts on $\mathcal{X}$ by permutation of components:*

$$(\sigma x)^{(k)} := x^{(\sigma^{-1}(k))}, \qquad \sigma \in S_K.$$

*Assume:*

1. *$p_0$ is permutation-invariant: $\sigma x_0 \overset{d}{=} x_0$ for all $\sigma \in S_K$;*

2. *$p_1$ is permutation-invariant: $\sigma x_1 \overset{d}{=} x_1$ for all $\sigma \in S_K$;*

3. *the vector field $v_\theta : \mathcal{X} \to \mathcal{X}$ is permutation-equivariant at $t = 0$:*

$$v_\theta(\sigma x, 0) = \sigma v_\theta(x, 0), \qquad \forall \sigma \in S_K.$$

*Define for $x_0, x_1 \in \mathcal{X}$ and $\sigma \in S_K$,*

$$\mathcal{L}(\sigma; x_0, x_1) := \left\| v_\theta(x_0, 0) - (\sigma x_1 - x_0) \right\|^2,$$

*and let*

$$\sigma(x_0, x_1, \theta) \in \arg\min_{\sigma \in S_K} \mathcal{L}(\sigma; x_0, x_1),$$

*assumed unique $(p_0 \otimes p_1)$-a.s. Define the permuted endpoint*

$$x_1^\sigma := \sigma(x_0, x_1, \theta)\, x_1.$$

*Then*

$$x_1^\sigma \sim p_1.$$

*Proof.* We split the argument into three steps.

**Step 1: Equivariance of the matching loss.** Fix $\tau \in S_K$. For any $\sigma \in S_K$ and $x_0, x_1 \in \mathcal{X}$,

$$\begin{aligned}
\mathcal{L}(\sigma; \tau x_0, x_1) &= \left\| v_\theta(\tau x_0, 0) - (\sigma x_1 - \tau x_0) \right\|^2 \\
&= \left\| \tau v_\theta(x_0, 0) - (\sigma x_1 - \tau x_0) \right\|^2 \\
&= \left\| \tau \left[ v_\theta(x_0, 0) - (\tau^{-1}\sigma x_1 - x_0) \right] \right\|^2 \\
&= \left\| v_\theta(x_0, 0) - ((\tau^{-1}\sigma)x_1 - x_0) \right\|^2 \\
&= \mathcal{L}(\tau^{-1}\sigma; x_0, x_1),
\end{aligned}$$

where we used permutation equivariance of $v_\theta$ and invariance of the norm under permutations. Consequently,

$$\sigma(\tau x_0, x_1, \theta) = \tau\, \sigma(x_0, x_1, \theta). \tag{44}$$

**Step 2: Conditional uniformity of $\sigma(x_0, x_1, \theta) \mid x_1$.** Fix $x_1 \in \mathcal{X}$ and define

$$A_\sigma(x_1) := \{ x_0 \in \mathcal{X} : \sigma(x_0, x_1, \theta) = \sigma \}.$$

From (44), we have $A_\sigma(x_1) = \sigma A_{\mathrm{id}}(x_1)$. By independence of $x_0$ and $x_1$ and permutation invariance of $p_0$,

$$\mathbb{P}\big(\sigma(x_0, x_1, \theta) = \sigma \mid x_1\big) = \mathbb{P}(x_0 \in A_\sigma(x_1)) = \mathbb{P}(x_0 \in A_{\mathrm{id}}(x_1)).$$

Since $\{A_\sigma(x_1)\}_{\sigma \in S_K}$ forms a partition of $\mathcal{X}$,

$$\mathbb{P}\big(\sigma(x_0, x_1, \theta) = \sigma \mid x_1\big) = \frac{1}{K!}, \qquad \forall \sigma \in S_K. \tag{A.2}$$

**Step 3: Marginal law of $x_1^\sigma$.** Let $B \subset \mathcal{X}$ be measurable. Conditioning on $x_1$,

$$\begin{aligned}
\mathbb{P}(x_1^\sigma \in B) &= \mathbb{E}\big[\mathbb{P}\big(\sigma(x_0, x_1, \theta)x_1 \in B \mid x_1\big)\big] \\
&= \mathbb{E}\left[ \frac{1}{K!} \sum_{\sigma \in S_K} \mathbf{1}\{\sigma x_1 \in B\} \right].
\end{aligned}$$

By permutation invariance of $p_1$, $\sigma x_1 \overset{d}{=} x_1$ for all $\sigma$, hence

$$\mathbb{P}(x_1^\sigma \in B) = \mathbb{P}(x_1 \in B).$$

This holds for all measurable $B$, so $x_1^\sigma \sim p_1$. $\qquad\square$

## C.2. Analysis of Population Losses

Here we provide insights into the population losses using the sample-wise alignment convention from Section 4.1. The teacher estimates $\bar{x}^{(k)}$ are understood to be the raw teacher outputs relabelled by a PIT matching to the latent source tuple $x$. This convention is used only to define source-wise errors such as $e^{(k)} = \bar{x}^{(k)} - x^{(k)}$; it is not an algorithmic step and does not impose a fixed ordering of sources across samples.

Recall that $x = [x^{(1)\top} \cdots x^{(K)\top}]^\top \in \mathbb{R}^{K \times d}$ denotes the true sources and $\bar{x} = [\bar{x}^{(1)\top} \cdots \bar{x}^{(K)\top}]^\top \in \mathbb{R}^{K \times d}$ their teacher estimates. In the population case, we associate $(K-1)$ independent "background" teacher sources to each estimate $x^{(k)}$ which follow

$$\bar{x}_{\mathrm{bg}}^{(k,2)}, \ldots, \bar{x}_{\mathrm{bg}}^{(k,K)} \overset{\mathrm{i.i.d.}}{\sim} \bar{p}_{\theta_\mathcal{T}}, \tag{45}$$

for

$$\bar{p}_{\theta_\mathcal{T}}(\bar{x}^{(k)}) = \int p_{\theta_\mathcal{T}}(\bar{x}^{(k)}, \bar{x}^{(-k)} \mid m) \, p(m) \, \mathrm{d}\bar{x}^{(-k)} \, \mathrm{d}m. \tag{46}$$

We write

$$\bar{x}_{\mathrm{bg}}^{(k)} = \left[(\bar{x}_{\mathrm{bg}}^{(k,2)})^\top, \ldots, (\bar{x}_{\mathrm{bg}}^{(k,K)})^\top\right]^\top. \tag{47}$$

We then obtain the source matrix anchored to source $k$

$$\tilde{x}_k = \left[(\bar{x}^{(k)})^\top, (\bar{x}_{\mathrm{bg}}^{(k,2)})^\top, \ldots, (\bar{x}_{\mathrm{bg}}^{(k,K)})^\top\right]^\top \in \mathbb{R}^{K \times d} \tag{48}$$

and the corresponding synthetic mixture

$$\tilde{m}_k := \mathbb{1}^\top \tilde{x}_k = \bar{x}^{(k)} + \sum_{j=2}^{K} \bar{x}_{\mathrm{bg}}^{(k,j)}. \tag{49}$$

### C.2.1. REGRESSION CASE

We assume throughout that the true sources, teacher estimates, and student predictions have finite second moments. Furthermore, for the gradient analysis, we assume $\mathbb{E}[\|\nabla_\theta \delta_\theta^{(k)}\|_F^2] < \infty$. For the student, the separator output for a pseudo-mixture $\tilde{m}_k$ is an unordered $K$-tuple, denoted $F_\theta(\tilde{m}_k) \in \mathbb{R}^{K \times d}$. Let

$$\rho_k \in \arg\min_{\rho \in S_K} \|F_\theta(\tilde{m}_k) - \rho \tilde{x}_k\|^2$$

be the PIT routing with respect to the source matrix $\tilde{x}_k$. We write

$$\hat{x}_k^\theta(\tilde{m}_k) := e_1^\top \rho_k^{-1} F_\theta(\tilde{m}_k) \in \mathbb{R}^{1 \times d}$$

for the student output row routed to the anchor teacher source $\bar{x}^{(k)}$, i.e. to the first row of $\tilde{x}_k$. Thus $\hat{x}_k^\theta(\tilde{m}_k)$ is not a fixed raw output index; it is the PIT-routed estimate assigned to the anchor. We recall that the artificial background sources used for different anchors are sampled independently across $k$.

In this case, we can simply rewrite the ReMixIT population loss as

$$\mathcal{L}_{\mathrm{RM}}(\theta) = \frac{1}{K} \sum_{k=1}^{K} \mathbb{E}\left[\left\|\hat{x}_k^\theta(\tilde{m}_k) - \bar{x}^{(k)}\right\|^2\right] = \mathbb{E}\left[\left\|\hat{x}_1^\theta(\tilde{m}_1) - \bar{x}^{(1)}\right\|^2\right], \tag{50}$$

and the (normalized) Self-Remixing population loss as

$$\mathcal{L}_{\mathrm{SR}}(\theta) = \frac{1}{K} \mathbb{E}\left[\left\|\left(\sum_{k=1}^{K} \hat{x}_k^\theta(\tilde{m}_k)\right) - m\right\|^2\right] \tag{51}$$

Using $\boldsymbol{m} = \sum_{k=1}^{K} \boldsymbol{x}^{(k)}$ and teacher mixture consistency $\boldsymbol{m} = \sum_{k=1}^{K} \bar{\boldsymbol{x}}^{(k)}$, we have

$$\mathcal{L}_{\text{SR}}(\theta) = \frac{1}{K} \mathbb{E} \left[ \left\| \sum_{k=1}^{K} \left( \hat{\boldsymbol{x}}_k^\theta (\tilde{\boldsymbol{m}}_k) - \boldsymbol{x}^{(k)} \right) \right\|^2 \right] \tag{52}$$

$$= \frac{1}{K} \mathbb{E} \left[ \left\| \sum_{k=1}^{K} \left( \hat{\boldsymbol{x}}_k^\theta (\tilde{\boldsymbol{m}}_k) - \bar{\boldsymbol{x}}^{(k)} \right) \right\|^2 \right]. \tag{53}$$

Finally we define the pseudo-supervised loss

$$\mathcal{L}_{\text{Sup}}(\theta) := \frac{1}{K} \sum_{k=1}^{K} \mathbb{E} \left[ \left\| \hat{\boldsymbol{x}}_k^\theta (\tilde{\boldsymbol{m}}_k) - \boldsymbol{x}^{(k)} \right\|^2 \right] = \mathbb{E} \left[ \left\| \hat{\boldsymbol{x}}_1^\theta (\tilde{\boldsymbol{m}}_1) - \boldsymbol{x}^{(1)} \right\|^2 \right]. \tag{54}$$

We call this a pseudo-supervised loss as $\tilde{\boldsymbol{m}}_k$ is not from the data distribution $p(\boldsymbol{m})$.

For each $k$, define the teacher-referenced error

$$\epsilon_\theta^{(k)} := \hat{\boldsymbol{x}}_k^\theta (\tilde{\boldsymbol{m}}_k) - \bar{\boldsymbol{x}}^{(k)}, \tag{55}$$

and true-reference error

$$\Delta_\theta^{(k)} := \hat{\boldsymbol{x}}_k^\theta (\tilde{\boldsymbol{m}}_k) - \boldsymbol{x}^{(k)}. \tag{56}$$

Define the teacher error on source $k$ as

$$e^{(k)} := \bar{\boldsymbol{x}}^{(k)} - \boldsymbol{x}^{(k)}. \tag{57}$$

Then

$$\Delta_\theta^{(k)} = \epsilon_\theta^{(k)} + e^{(k)} \qquad \Longleftrightarrow \qquad \epsilon_\theta^{(k)} = \Delta_\theta^{(k)} - e^{(k)}. \tag{58}$$

Let $\mathcal{A}_k := \sigma(\boldsymbol{x}^{(k)}, \bar{\boldsymbol{x}}^{(k)})$ denote the anchor information, i.e. the true source and its teacher estimate. The background-averaged errors are

$$\mu_\theta^{(k)} := \mathbb{E} \left[ \epsilon_\theta^{(k)} \mid \mathcal{A}_k \right], \tag{59}$$

and

$$\delta_\theta^{(k)} := \mathbb{E} \left[ \Delta_\theta^{(k)} \mid \mathcal{A}_k \right]. \tag{60}$$

Finally the teacher MSE level is

$$\sigma_{\mathcal{T}}^2 := \mathbb{E} \left[ \left\| e^{(1)} \right\|^2 \right], \tag{61}$$

where the expectation is over the joint law of the true source, mixture, and teacher estimate.

The following result establishes useful expressions and relationships between those losses.

**Proposition C.2** (Relationships between the losses). *The ReMixIT, Self-remixing and pseudo-supervised losses satisfy the following relationships*

$$\mathcal{L}_{\text{RM}}(\theta) = \mathcal{L}_{\text{Sup}}(\theta) + \sigma_{\mathcal{T}}^2 - 2 \mathbb{E} \left[ \left\langle e^{(1)}, \delta_\theta^{(1)} \right\rangle \right], \tag{62}$$

$$\mathcal{L}_{\text{SR}}(\theta) = \mathcal{L}_{\text{Sup}}(\theta) + (K-1) \mathbb{E} \left[ \left\langle \delta_\theta^{(1)}, \delta_\theta^{(2)} \right\rangle \right], \tag{63}$$

*and*

$$\mathcal{L}_{\text{SR}}(\theta) = \mathcal{L}_{\text{RM}}(\theta) + (K-1) \mathbb{E} \left[ \left\langle \mu_\theta^{(1)}, \mu_\theta^{(2)} \right\rangle \right], \tag{64}$$

*Proof.* To prove (62), we use (58) for $k = 1$ gives $\epsilon_\theta^{(1)} = \Delta_\theta^{(1)} - e^{(1)}$, hence

$$\mathcal{L}_{\mathrm{RM}}(\theta) = \mathbb{E}\big[\big\|\epsilon_\theta^{(1)}\big\|^2\big] = \mathbb{E}\big[\big\|\Delta_\theta^{(1)} - e^{(1)}\big\|^2\big] = \mathbb{E}\big[\big\|\Delta_\theta^{(1)}\big\|^2\big] + \mathbb{E}\big[\big\|e^{(1)}\big\|^2\big] - 2\mathbb{E}[\big\langle e^{(1)}, \Delta_\theta^{(1)}\big\rangle].$$

The first term equals $\mathcal{L}_{\mathrm{Sup}}(\theta)$ by (54), and the second equals $\sigma_\mathcal{T}^2$ by (61). We define for each $k$, the $\sigma$-field $\mathcal{A}_k = \sigma(\boldsymbol{x}^k, \bar{\boldsymbol{x}}^k)$. Using this definition, the last term can be re-expressed as

$$\mathbb{E}[\big\langle e^{(1)}, \Delta_\theta^{(1)}\big\rangle] = \mathbb{E}[\mathbb{E}[\big\langle e^{(1)}, \Delta_\theta^{(1)}\big\rangle | \mathcal{A}_1]] = \mathbb{E}[\big\langle e^{(1)}, \delta_\theta^{(1)}\big\rangle]. \tag{65}$$

To prove (63), we combine (52) and (56) to obtain

$$\mathcal{L}_{\mathrm{SR}}(\theta) = \frac{1}{K}\mathbb{E}\Big[\Big\|\sum_{k=1}^K \Delta_\theta^{(k)}\Big\|^2\Big] = \frac{1}{K}\sum_{k=1}^K \mathbb{E}\big[\big\|\Delta_\theta^{(k)}\big\|^2\big] + \frac{1}{K}\sum_{k\neq\ell}\mathbb{E}\big[\big\langle\Delta_\theta^{(k)}, \Delta_\theta^{(\ell)}\big\rangle\big] \tag{66}$$

$$= \mathbb{E}\big[\big\|\Delta_\theta^{(1)}\big\|^2\big] + (K-1)\mathbb{E}\big[\big\langle\Delta_\theta^{(1)}, \Delta_\theta^{(2)}\big\rangle\big], \tag{67}$$

where we used that $\mathbb{E}\big[\big\|\Delta_\theta^{(k)}\big\|^2\big] = \mathbb{E}\big[\big\|\Delta_\theta^{(1)}\big\|^2\big]$ and $\mathbb{E}\big[\big\langle\Delta_\theta^{(k)}, \Delta_\theta^{(\ell)}\big\rangle\big] = \mathbb{E}\big[\big\langle\Delta_\theta^{(1)}, \Delta_\theta^{(2)}\big\rangle\big]$ for any $k, \ell$ $(k \neq \ell)$ thanks to exchangeability.

Let $k \neq \ell$. Conditionally on $\mathcal{A}_k \vee \mathcal{A}_\ell$, the only remaining randomness in $\Delta_\theta^{(k)}$ and $\Delta_\theta^{(\ell)}$ comes from the independently sampled artificial background blocks used to construct $\tilde{\boldsymbol{m}}_k$ and $\tilde{\boldsymbol{m}}_\ell$. Hence $\Delta_\theta^{(k)}$ and $\Delta_\theta^{(\ell)}$ are conditionally independent given $\mathcal{A}_k \vee \mathcal{A}_\ell$, and therefore

$$\mathbb{E}[\langle\Delta_\theta^{(k)}, \Delta_\theta^{(\ell)}\rangle | \mathcal{A}_k \vee \mathcal{A}_\ell] = \big\langle\mathbb{E}[\Delta_\theta^{(k)} | \mathcal{A}_k \vee \mathcal{A}_\ell], \mathbb{E}[\Delta_\theta^{(\ell)} | \mathcal{A}_k \vee \mathcal{A}_\ell]\big\rangle. \tag{68}$$

Moreover, since the background block for anchor $k$ is independent of $\mathcal{A}_\ell$ conditional on $\mathcal{A}_k$,

$$\mathbb{E}[\Delta_\theta^{(k)} | \mathcal{A}_k \vee \mathcal{A}_\ell] = \mathbb{E}[\Delta_\theta^{(k)} | \mathcal{A}_k] = \delta_\theta^{(k)},$$

and similarly

$$\mathbb{E}[\Delta_\theta^{(\ell)} | \mathcal{A}_k \vee \mathcal{A}_\ell] = \delta_\theta^{(\ell)}.$$

Thus

$$\mathbb{E}[\langle\Delta_\theta^{(k)}, \Delta_\theta^{(\ell)}\rangle | \mathcal{A}_k \vee \mathcal{A}_\ell] = \langle\delta_\theta^{(k)}, \delta_\theta^{(\ell)}\rangle.$$

We prove (64) identically by combining (53) and (55). □

The expression (62) shows that the ReMixIT loss deviates from (pseudo)-supervised learning by a term $-2\mathbb{E}\big[\big\langle e^{(1)}, \delta_\theta^{(1)}\big\rangle\big]$ which depends on the teacher error $e^{(1)}$ and how the averaged (over the "background" sources) student error aligns with it. Note that a similar expression appeared in (Tzinis et al., 2022) for finite $B$ but it is claimed incorrectly therein that the term equivalent to $\mathbb{E}\big[\big\langle e^{(1)}, \delta_\theta^{(1)}\big\rangle\big]$ in their decomposition is independent of $\theta$ as $B \to \infty$. However, we can expect this term to be of moderate magnitude compared to the pseudo-supervised loss. Indeed, although the student is initialized at the teacher parameters, the student is never evaluated on the same inputs (and input distribution) as the teacher. Due to background remixing, the student sees mixtures of the form $\tilde{\boldsymbol{m}}_1 = \tilde{\boldsymbol{x}}_1 +$ random background sources, where the background sources are independent of the original mixture. So even at initialization the student's output varies with the background sources; averaging over this variability produces thus a smoothed estimate that suppresses mixture-specific teacher error. As a result, we expect the remix-averaged student error $\delta_\theta^{(1)}$ to be weakly correlated with the teacher error.

The expression (63) shows instead that Self-Remixing only deviates from (pseudo)-supervised learning by the cross correlation of the student errors $(K-1)\mathbb{E}\big[\big\langle\delta_\theta^{(1)}, \delta^{(2)}\big\rangle\big]$. When this coupling term is small; e.g., because the background-averaged errors $\delta_\theta^{(1)}, \delta_\theta^{(2)}$ are small or weakly correlated across sources, $\mathcal{L}_{\mathrm{SR}}(\theta)$ primarily reduces $\mathcal{L}_{\mathrm{Sup}}(\theta)$, even though the objective does not explicitly penalize per-source errors.

We now study by how much the gradients of the ReMixIT and Self-remixing losses differ from the gradient of the pseudo-supervised loss.

**Proposition C.3.** *Assume that $\theta \mapsto \hat{x}_{\theta,k}(\tilde{m}_k)$ is differentiable and that we can interchange $\nabla_\theta$ with the relevant expectations/conditional expectations. We identify row signals in $\mathbb{R}^{1 \times d}$ with vectors in $\mathbb{R}^d$; if $\theta \in \mathbb{R}^p$, then $\nabla_\theta \delta_\theta^{(k)} \in \mathbb{R}^{d \times p}$ and $(\nabla_\theta \delta_\theta^{(k)})^\top e^{(k)} \in \mathbb{R}^p$.*

*The gradient of the ReMixIT and Self-remixing loss satisfy*

$$\nabla_\theta \mathcal{L}_{\mathrm{RM}}(\theta) - \nabla_\theta \mathcal{L}_{\mathrm{Sup}}(\theta) = -2 \, \mathbb{E}\left[ \left( \nabla_\theta \delta_\theta^{(1)} \right)^\top e^{(1)} \right], \tag{69}$$

$$\nabla_\theta \mathcal{L}_{\mathrm{SR}}(\theta) - \nabla_\theta \mathcal{L}_{\mathrm{Sup}}(\theta) = (K-1) \, \mathbb{E}\left[ \left( \nabla_\theta \delta_\theta^{(1)} \right)^\top \delta_\theta^{(2)} + \left( \nabla_\theta \delta_\theta^{(2)} \right)^\top \delta_\theta^{(1)} \right]. \tag{70}$$

*By exchangeability of sources, the right-hand side of (70) can also be written as*

$$\nabla_\theta \mathcal{L}_{\mathrm{SR}}(\theta) - \nabla_\theta L_{\mathrm{Sup}}(\theta) = 2(K-1) \, \mathbb{E}\left[ \left( \nabla_\theta \delta_\theta^{(1)} \right)^\top \delta_\theta^{(2)} \right]. \tag{71}$$

*Let $\|\cdot\|$ denote the Euclidean norm on signals and $\|\cdot\|_F$ the Frobenius norm on Jacobians. Then*

$$\|\nabla_\theta \mathcal{L}_{\mathrm{RM}}(\theta) - \nabla_\theta \mathcal{L}_{\mathrm{Sup}}(\theta)\| \le 2 \, \mathbb{E}\left[ \|e^{(1)}\| \, \|\nabla_\theta \delta_\theta^{(1)}\|_F \right] \le 2 \, \sigma_{\mathcal{T}} \sqrt{\mathbb{E}\|\nabla_\theta \delta_\theta^{(1)}\|_F^2}, \tag{72}$$

$$\|\nabla_\theta \mathcal{L}_{\mathrm{SR}}(\theta) - \nabla_\theta \mathcal{L}_{\mathrm{Sup}}(\theta)\| \le 2(K-1) \, \mathbb{E}\left[ \|\delta_\theta^{(2)}\| \, \|\nabla_\theta \delta_\theta^{(1)}\|_F \right] \le 2(K-1) \sqrt{\mathbb{E}\|\delta_\theta^{(1)}\|^2} \sqrt{\mathbb{E}\|\nabla_\theta \delta_\theta^{(1)}\|_F^2}. \tag{73}$$

*In particular, the Self-Remixing gradient deviates from the pseudo-supervised gradient in proportion to the* background-averaged *student error magnitude $\sqrt{\mathbb{E}\|\delta_\theta^{(1)}\|^2}$.*

*Proof.* Differentiate (62) w.r.t. $\theta$. The term $\sigma_{\mathcal{T}}^2$ is constant in $\theta$ (teacher fixed), so

$$\nabla_\theta \mathcal{L}_{\mathrm{RM}}(\theta) - \nabla_\theta \mathcal{L}_{\mathrm{Sup}}(\theta) = -2 \, \nabla_\theta \mathbb{E}\langle e^{(1)}, \delta_\theta^{(1)} \rangle.$$

Under the stated interchange assumptions and since $e^{(1)}$ does not depend on $\theta$,

$$\nabla_\theta \mathbb{E}\langle e^{(1)}, \delta_\theta^{(1)} \rangle = \mathbb{E}\left[ (\nabla_\theta \delta_\theta^{(1)})^\top e^{(1)} \right],$$

which gives (69). Similarly differentiating (63) yields (70), and exchangeability implies (71).

Apply $\|\mathbb{E}Z\| \le \mathbb{E}\|Z\|$ to (69) and use $\|(\nabla_\theta \delta)^\top e\| \le \|\nabla_\theta \delta\|_F \|e\|$ to get the first inequality in (72). The second follows from Cauchy–Schwarz. The Self-Remixing bound (73) follows the same way from (71) and exchangeability of $\delta_\theta^{(1)}$ and $\delta_\theta^{(2)}$. $\qquad \square$

### C.2.2. GENERATIVE CASE

We now rewrite the population ReMixIT and Self-Remixing *flow-matching* objectives of Section 4.1. Throughout we work in the population regime ($B \to \infty$) and use the same sample-wise routing convention as in the regression case. Teacher estimates are already relabelled by the teacher-side alignment convention, and student FM outputs are PIT-routed with respect to the matrix $\tilde{x}_k$ composed of the teacher source estimate and $K-1$ i.i.d. background sources. After applying this routing, we suppress the explicit permutation notation and treat the displayed quantities as already aligned.

Recall that we sample $\tilde{x}_{k,0} \sim p_0(\cdot \mid \tilde{m}_k)$ and $t \sim \mathcal{U}[0,1]$, and define the linear interpolant

$$\tilde{x}_{k,t} := (1-t)\tilde{x}_{k,0} + t\tilde{x}_k. \tag{74}$$

Given a student velocity $v_\theta(\cdot, t, \tilde{m}_k) \in \mathbb{R}^{K \times d}$, we also define the associated time-$t$ *clean-source estimate* by

$$\hat{x}_{k,1,\theta}(\tilde{x}_{k,t}, t, \tilde{m}_k) := \tilde{x}_{k,t} + (1-t) \, v_\theta(\tilde{x}_{k,t}, t, \tilde{m}_k) \in \mathbb{R}^{K \times d}. \tag{75}$$

It follows that

$$\hat{x}_{k,1,\theta}(\tilde{x}_{k,t}, t, \tilde{m}_k) - \tilde{x}_k = (1-t)\Big( v_\theta(\tilde{x}_{k,t}, t, \tilde{m}_k) - (\tilde{x}_k - \tilde{x}_{k,0}) \Big). \tag{76}$$

**Population ReMixIT–Flow loss.**    The population ReMixIT–Flow objective is

$$L_{\text{RM-FM}}^{\infty}(\theta) := \frac{1}{K}\mathbb{E}_{t\sim\lambda}\Big[\big\|\boldsymbol{v}_\theta(\tilde{\boldsymbol{x}}_{k,t}, t, \tilde{\boldsymbol{m}}_k) - (\tilde{\boldsymbol{x}}_k - \tilde{\boldsymbol{x}}_{k,0})\big\|_F^2\Big] \tag{77}$$

where $t \sim \lambda$. Using (76), this is equivalently a (reweighted) regression loss on the time-dependent clean estimate:

$$L_{\text{RM-FM}}^{\infty}(\theta) = \frac{1}{K}\mathbb{E}_{t\sim\mathcal{U}[0,1]}\Big[\beta_t\big\|\hat{\boldsymbol{x}}_{k,1,\theta}(\tilde{\boldsymbol{x}}_{k,t}, t, \tilde{\boldsymbol{m}}_k) - \tilde{\boldsymbol{x}}_k\big\|_F^2\Big] \tag{78}$$

where $\beta_t = \lambda_t/(1-t)^2$.

**Population Self-Remixing–Flow loss.**    Self-Remixing supervises against the observed mixture $\boldsymbol{m} = \sum_{k=1}^{K} \boldsymbol{x}^{(k)}$, and in the population construction the anchor source $\bar{\boldsymbol{x}}^{(k)}$ occupies the first row of $\tilde{\boldsymbol{x}}_k$ by definition. Let $e_1 \in \mathbb{R}^K$ denote the first standard basis vector. Define the induced time-$t$ mixture estimate

$$\hat{\boldsymbol{m}}_\theta(t) := e_1^\top \sum_{k=1}^{K} \hat{\boldsymbol{x}}_{k,1,\theta}(\tilde{\boldsymbol{x}}_{k,t}, t, \tilde{\boldsymbol{m}}_k) \in \mathbb{R}^{1\times d}. \tag{79}$$

Define the per-anchor drift residual routed to the anchor component by

$$\boldsymbol{r}_{k,\theta}(t) := e_1^\top\Big(\boldsymbol{v}_\theta(\tilde{\boldsymbol{x}}_{k,t}, t, \tilde{\boldsymbol{m}}_k) - (\tilde{\boldsymbol{x}}_k - \tilde{\boldsymbol{x}}_{k,0})\Big). \tag{80}$$

Assume here, as in the regression analysis, exact teacher mixture consistency $\sum_{k=1}^{K} \bar{\boldsymbol{x}}^{(k)} = \sum_{k=1}^{K} \boldsymbol{x}^{(k)} = \boldsymbol{m}$. Then, as in Section 4.2, we have

$$\hat{\boldsymbol{m}}_\theta(t) - \boldsymbol{m} = (1-t)\sum_{k=1}^{K} \boldsymbol{r}_{k,\theta}(t), \tag{81}$$

and the population Self-Remixing–Flow objective (renormalized here by $1/K$) is

$$L_{\text{SR-FM}}^{\infty}(\theta) := \frac{1}{K}\mathbb{E}_{t\sim\lambda}\Big[\big\|\sum_{k=1}^{K}\boldsymbol{r}_{k,\theta}(t)\big\|^2\Big] = \frac{1}{K}\mathbb{E}_{t\sim\mathcal{U}[0,1]}\Big[\beta_t\big\|\hat{\boldsymbol{m}}_\theta(t) - \boldsymbol{m}\big\|^2\Big]. \tag{82}$$

Equations (78) and (82) show that both unsupervised FM objectives can be interpreted as (reweighted) regression losses on the time-dependent clean-source estimate $\hat{\boldsymbol{x}}_{k,1,\theta}(\tilde{\boldsymbol{x}}_{k,t}, t, \tilde{\boldsymbol{m}}_k)$.

We now derive FM counterparts of Propositions C.2-C.3 by working with the time-dependent clean estimate $\hat{\boldsymbol{x}}_{k,1,\theta}(\tilde{\boldsymbol{x}}_{k,t}, t, \tilde{\boldsymbol{m}}_k)$ and projecting to the *anchor component* (first row). Define

$$\hat{\boldsymbol{x}}_{\theta,t}^{(k)} := e_1^\top\hat{\boldsymbol{x}}_{k,1,\theta}(\tilde{\boldsymbol{x}}_{k,t}, t, \tilde{\boldsymbol{m}}_k), \qquad \bar{\boldsymbol{x}}^{(k)} := e_1^\top\tilde{\boldsymbol{x}}_k, \qquad \boldsymbol{x}^{(k)} := \boldsymbol{x}_{k,\text{true}}.$$

Define the pseudo-supervised FM loss (anchor component) by

$$L_{\text{Sup-FM}}^{\infty}(\theta) := \mathbb{E}_{t\sim\mathcal{U}[0,1]}\Big[\beta_t\big\|\hat{\boldsymbol{x}}_{\theta,t}^{(1)} - \boldsymbol{x}^{(1)}\big\|^2\Big], \tag{83}$$

and the anchor-projected ReMixIT–Flow loss

$$L_{\text{RM-FM}}^{\infty}(\theta) := \mathbb{E}_{t\sim\mathcal{U}[0,1]}\Big[\beta_t\big\|\hat{\boldsymbol{x}}_{\theta,t}^{(1)} - \bar{\boldsymbol{x}}^{(1)}\big\|^2\Big]. \tag{84}$$

**Errors and background-averaged errors.**    Define teacher error and true/teacher-referenced student errors (anchor component):

$$e^{(k)} := \bar{\boldsymbol{x}}^{(k)} - \boldsymbol{x}^{(k)}, \tag{85}$$

$$\epsilon_{\theta,t}^{(k)} := \hat{\boldsymbol{x}}_{\theta,t}^{(k)} - \bar{\boldsymbol{x}}^{(k)}, \tag{86}$$

$$\Delta_{\theta,t}^{(k)} := \hat{\boldsymbol{x}}_{\theta,t}^{(k)} - \boldsymbol{x}^{(k)}. \tag{87}$$

Then $\Delta_{\theta,t}^{(k)} = \epsilon_{\theta,t}^{(k)} + e^{(k)}$. We define for each $k$, the $\sigma$-field $\mathcal{A}_k = \sigma(\boldsymbol{x}^{(k)}, \bar{\boldsymbol{x}}^{(k)}, t)$. This represents the information contained in the $k$-th anchor source, its teacher estimate, and the time step, averaging out the randomness of the background sources and the flow matching noise. Define the background-averaged (systematic) errors

$$\mu_{\theta,t}^{(k)} := \mathbb{E}[\epsilon_{\theta,t}^{(k)} \mid \mathcal{A}_k], \qquad \delta_{\theta,t}^{(k)} := \mathbb{E}[\Delta_{\theta,t}^{(k)} \mid \mathcal{A}_k],$$

and $\sigma_{\mathcal{T}}^2 := \mathbb{E}\|e^{(1)}\|^2$.

**Proposition C.4** (FM analogues of Propositions C.2-C.3). *Assume the PIT alignment and the population construction above, independent artificial backgrounds across anchors, exact teacher mixture consistency, and that $\lambda = (\lambda_t)_{t\in[0,1]}$ is selected such that all the following expectations are finite. Then:*

1. *(**Loss identities**) We have*

$$L_{\text{RM-FM}}^{\infty}(\theta) = L_{\text{Sup-FM}}^{\infty}(\theta) + \mathbb{E}\left[\beta_t \|e^{(1)}\|^2\right] - 2\,\mathbb{E}\left[\beta_t \langle e^{(1)}, \delta_{\theta,t}^{(1)}\rangle\right], \tag{88}$$

$$L_{\text{SR-FM}}^{\infty}(\theta) = L_{\text{Sup-FM}}^{\infty}(\theta) + (K-1)\,\mathbb{E}\left[\beta_t \langle \delta_{\theta,t}^{(1)}, \delta_{\theta,t}^{(2)}\rangle\right]. \tag{89}$$

2. *(**Gradient identities and bounds**) If $\theta \mapsto \hat{\boldsymbol{x}}_{\theta,t}^{(k)}$ is differentiable and gradients may be interchanged with expectations, then*

$$\nabla_\theta L_{\text{RM-FM}}^{\infty}(\theta) - \nabla_\theta L_{\text{Sup-FM}}^{\infty}(\theta) = -2\,\mathbb{E}\left[\beta_t (\nabla_\theta \delta_{\theta,t}^{(1)})^\top e^{(1)}\right], \tag{90}$$

$$\nabla_\theta L_{\text{SR-FM}}^{\infty}(\theta) - \nabla_\theta L_{\text{Sup-FM}}^{\infty}(\theta) = 2(K-1)\,\mathbb{E}\left[\beta_t (\nabla_\theta \delta_{\theta,t}^{(1)})^\top \delta_{\theta,t}^{(2)}\right], \tag{91}$$

*and consequently*

$$\left\|\nabla_\theta L_{\text{RM-FM}}^{\infty}(\theta) - \nabla_\theta L_{\text{Sup-FM}}^{\infty}(\theta)\right\| \le 2\,\mathbb{E}\left[\beta_t \|e^{(1)}\| \|\nabla_\theta \delta_{\theta,t}^{(1)}\|_F\right], \tag{92}$$

$$\left\|\nabla_\theta L_{\text{SR-FM}}^{\infty}(\theta) - \nabla_\theta L_{\text{Sup-FM}}^{\infty}(\theta)\right\| \le 2(K-1)\,\mathbb{E}\left[\beta_t \|\delta_{\theta,t}^{(2)}\| \|\nabla_\theta \delta_{\theta,t}^{(1)}\|_F\right]. \tag{93}$$

*Proof sketch.* All identities follow by the same algebra as Section C.2.1 applied to the reweighted clean-estimate losses. For (88), expand $\|\Delta_{\theta,t}^{(1)} - e^{(1)}\|^2$ inside (84) and use the tower property conditioning on $\mathcal{A}_1$ (where $e^{(1)}$ is measurable) to replace $\mathbb{E}\langle e^{(1)}, \Delta_{\theta,t}^{(1)}\rangle$ by $\mathbb{E}\langle e^{(1)}, \delta_{\theta,t}^{(1)}\rangle$. For (89), expand $\|\sum_k \Delta_{\theta,t}^{(k)}\|^2$ and condition on $\mathcal{A}_1 \vee \mathcal{A}_2$ to factorize cross terms through $\delta_{\theta,t}^{(k)}$, using independence of anchor backgrounds across $k$. Gradient identities follow by differentiating the loss identities; norm bounds are immediate from $\|\mathbb{E}Z\| \le \mathbb{E}\|Z\|$ and Cauchy–Schwarz. $\square$

## C.3. Alternative Derivation of the Self-Remixing Loss for Flow Matching

In Section 3.3, the cleaned data estimator used in the Self-Remixing Flow Matching objective is defined by (23) which we recall here

$$\widehat{\boldsymbol{X}}_{1,\theta}^{\boldsymbol{\Upsilon},\text{old}}(\widetilde{\boldsymbol{X}}_t^{\boldsymbol{\Upsilon}}, t, \widetilde{\boldsymbol{M}}) = \widetilde{\boldsymbol{X}}_t^{\boldsymbol{\Upsilon}} + (1-t)\boldsymbol{v}_\theta(\widetilde{\boldsymbol{X}}_t^{\boldsymbol{\Upsilon}}, t, \widetilde{\boldsymbol{M}}), \tag{94}$$

with $\widetilde{\boldsymbol{X}}_t^{\boldsymbol{\Upsilon}} = (1-t)\widetilde{\boldsymbol{X}}_0 + t\boldsymbol{\Upsilon}\widetilde{\boldsymbol{X}}_1$. We consider the following alternative velocity-residual endpoint proxy,

$$\widehat{\boldsymbol{X}}_{1,\theta}^{\boldsymbol{\Upsilon},\text{new}} := \widetilde{\boldsymbol{X}}_0 + \boldsymbol{v}_\theta(\widetilde{\boldsymbol{X}}_t^{\boldsymbol{\Upsilon}}, t, \widetilde{\boldsymbol{M}}), \tag{95}$$

which coincides with the supervised endpoint estimator at $t = 0$ but is not, for $t \neq 0$, the usual FM conditional denoiser $\widetilde{\boldsymbol{X}}_t^{\boldsymbol{\Upsilon}} + (1-t)\boldsymbol{v}_\theta(\widetilde{\boldsymbol{X}}_t^{\boldsymbol{\Upsilon}}, t, \widetilde{\boldsymbol{M}})$.

For $t \neq 1$, it is easy to check that the two estimators are related by a simple affine transformation. Eliminating the drift $\boldsymbol{v}_\theta$ from (94) yields

$$\boldsymbol{v}_\theta(\widetilde{\boldsymbol{X}}_t^{\boldsymbol{\Upsilon}}, t, \widetilde{\boldsymbol{M}}) = \frac{\widehat{\boldsymbol{X}}_{1,\theta}^{\boldsymbol{\Upsilon},\text{old}} - \widetilde{\boldsymbol{X}}_t^{\boldsymbol{\Upsilon}}}{1-t}, \tag{96}$$

and therefore

$$\widehat{\boldsymbol{X}}_{1,\theta}^{\boldsymbol{\Upsilon},\text{new}} = \widetilde{\boldsymbol{X}}_0 + \frac{\widehat{\boldsymbol{X}}_{1,\theta}^{\boldsymbol{\Upsilon},\text{old}} - \widetilde{\boldsymbol{X}}_t^{\boldsymbol{\Upsilon}}}{1-t} = \frac{\widehat{\boldsymbol{X}}_{1,\theta}^{\boldsymbol{\Upsilon},\text{old}} - t\,\boldsymbol{\Upsilon}\,\widetilde{\boldsymbol{X}}_1}{1-t}. \tag{97}$$

Conversely,

$$\widehat{\boldsymbol{X}}_{1,\theta}^{\boldsymbol{\Upsilon},\text{old}} = \widetilde{\boldsymbol{X}}_t^{\boldsymbol{\Upsilon}} + (1-t)\big( \widehat{\boldsymbol{X}}_{1,\theta}^{\boldsymbol{\Upsilon},\text{new}} - \widetilde{\boldsymbol{X}}_0 \big). \tag{98}$$

Recall that the Self-Remixing loss is defined as

$$\mathcal{L}(\theta) = \mathbb{E}\Big[\big\|(\boldsymbol{I}_B \otimes \mathbb{1}^\top)\boldsymbol{\Pi}^{-1}\big(\boldsymbol{\Upsilon}^{-1}\,\widehat{\boldsymbol{X}}_{1,\theta}^{\boldsymbol{\Upsilon}} - \widetilde{\boldsymbol{X}}_1\big)\big\|^2\Big] \tag{99}$$

So using (94), we get

$$\boldsymbol{\Upsilon}^{-1}\,\widehat{\boldsymbol{X}}_{1,\theta}^{\boldsymbol{\Upsilon},\text{old}} - \widetilde{\boldsymbol{X}}_1 = (1-t)\,\boldsymbol{\Upsilon}^{-1}\Big(\boldsymbol{v}_\theta(\widetilde{\boldsymbol{X}}_t^{\boldsymbol{\Upsilon}},t,\widetilde{\boldsymbol{M}}) - (\boldsymbol{\Upsilon}\,\widetilde{\boldsymbol{X}}_1 - \widetilde{\boldsymbol{X}}_0)\Big), \tag{100}$$

so that the corresponding loss is

$$\mathcal{L}_{\text{old}}(\theta) = \mathbb{E}\Big[(1-t)^2\big\|(\boldsymbol{I}_B \otimes \mathbb{1}^\top)\,\boldsymbol{\Pi}^{-1}\boldsymbol{\Upsilon}^{-1}\big(\boldsymbol{v}_\theta(\widetilde{\boldsymbol{X}}_t^{\boldsymbol{\Upsilon}},t,\widetilde{\boldsymbol{M}}) - (\boldsymbol{\Upsilon}\,\widetilde{\boldsymbol{X}}_1 - \widetilde{\boldsymbol{X}}_0)\big)\big\|^2\Big]. \tag{101}$$

In Section 3.3, the factor $(1-t)^2$ is dropped, leading to (28).

For the alternative estimator (95), we obtain instead

$$\boldsymbol{\Upsilon}^{-1}\,\widehat{\boldsymbol{X}}_{1,\theta}^{\boldsymbol{\Upsilon},\text{new}} - \widetilde{\boldsymbol{X}}_1 = \boldsymbol{\Upsilon}^{-1}\Big(\boldsymbol{v}_\theta(\widetilde{\boldsymbol{X}}_t^{\boldsymbol{\Upsilon}},t,\widetilde{\boldsymbol{M}}) - (\boldsymbol{\Upsilon}\,\widetilde{\boldsymbol{X}}_1 - \widetilde{\boldsymbol{X}}_0)\Big), \tag{102}$$

and hence the induced loss is

$$\mathcal{L}_{\text{new}}(\theta) = \mathbb{E}\Big[\big\|(\boldsymbol{I}_B \otimes \mathbb{1}^\top)\,\boldsymbol{\Pi}^{-1}\boldsymbol{\Upsilon}^{-1}\big(\boldsymbol{v}_\theta(\widetilde{\boldsymbol{X}}_t^{\boldsymbol{\Upsilon}},t,\widetilde{\boldsymbol{M}}) - (\boldsymbol{\Upsilon}\,\widetilde{\boldsymbol{X}}_1 - \widetilde{\boldsymbol{X}}_0)\big)\big\|^2\Big]. \tag{103}$$

**Comparison.** Equations (101) and (103) show that the two estimators induce the same residual, but with a different time-weighting:

$$\mathcal{L}_{\text{old}}(\theta) = \mathbb{E}\big[(1-t)^2 \cdot \ell(\theta,t)\big], \qquad \mathcal{L}_{\text{new}}(\theta) = \mathbb{E}\big[\ell(\theta,t)\big],$$

where $\ell(\theta,t)$ denotes the squared mixture residual. Thus, using the alternative endpoint proxy (95) is equivalent to removing the $(1-t)^2$ weighting inside the expectation and yields the reweighted objective (28).

# D. Experimental Details

## D.1. Further Algorithmic Details

---

**Algorithm 2** SURF: Unsupervised Remixing Flow

---

1: **Require:** Student $\theta$, Teacher $\theta_\mathcal{T}$, Batch $B$, EMA $\alpha$
2: **while** not converged **do**
3:     {**Phase 1: Generative Remixing**}
4:     $\boldsymbol{M} \sim p(\boldsymbol{M})$                                             {Draw observed mixtures from data}
5:     $\bar{\boldsymbol{X}} \sim p_{\theta_\mathcal{T}}(\bar{\boldsymbol{X}} \mid \boldsymbol{M})$         {Teacher generates source estimates}
6:     $\boldsymbol{\Pi} \sim \mathcal{U}(S_{BK})$         {Sample global permutation matrix}
7:     $\widetilde{\boldsymbol{X}}_1 = \boldsymbol{\Pi}\bar{\boldsymbol{X}}$         {Shuffle sources across the entire batch}
8:     $\widetilde{\boldsymbol{M}} = (\boldsymbol{I}_B \otimes \mathbb{1}^\top)\, \widetilde{\boldsymbol{X}}_1$     {Construct remixed synthetic mixtures}
9:
10:     {**Phase 2: Flow Matching Path**}
11:     $t \sim \mathcal{U}(0,1)$         {Sample time for the probability path}
12:     $\text{vec}(\boldsymbol{Z}) \sim \mathcal{N}(\boldsymbol{0}, \boldsymbol{I}_{BKd})$ for $\boldsymbol{Z} \in \mathbb{R}^{BK \times d}$     {Sample standard Gaussian noise}
13:     $\widetilde{\boldsymbol{X}}_0 = \frac{1}{K}(\boldsymbol{I}_B \otimes \mathbb{1})\, \widetilde{\boldsymbol{M}} + (\boldsymbol{I}_B \otimes \boldsymbol{P}^\perp)\boldsymbol{Z}$     {Project matrix-valued noise for consistency}
14:     $\boldsymbol{\Upsilon} = \arg\min_{\boldsymbol{\Gamma}} \|\boldsymbol{v}_\theta(\widetilde{\boldsymbol{X}}_0, 0, \widetilde{\boldsymbol{M}}) - (\boldsymbol{\Gamma}\, \widetilde{\boldsymbol{X}}_1 - \widetilde{\boldsymbol{X}}_0)\|^2$     {PIT Alignment}
15:     $\widetilde{\boldsymbol{X}}_t^{\boldsymbol{\Upsilon}} = (1-t)\, \widetilde{\boldsymbol{X}}_0 + t\boldsymbol{\Upsilon}\, \widetilde{\boldsymbol{X}}_1$     {Conditional flow interpolant}
16:
17:     {**Phase 3: Objectives & Update**}
18:     $\boldsymbol{R}_t := \boldsymbol{v}_\theta(\widetilde{\boldsymbol{X}}_t^{\boldsymbol{\Upsilon}}, t, \widetilde{\boldsymbol{M}}) - (\boldsymbol{\Upsilon}\, \widetilde{\boldsymbol{X}}_1 - \widetilde{\boldsymbol{X}}_0)$     {Drift residual}
19:     $\mathcal{L}_{\text{RM-FM}} = \|\boldsymbol{R}_t\|^2$     {ReMixIT Loss}
20:     $\mathcal{L}_{\text{SR-FM}} = \|(\boldsymbol{I}_B \otimes \mathbb{1}^\top)\boldsymbol{\Pi}^{-1}\boldsymbol{\Upsilon}^{-1}\boldsymbol{R}_t\|^2$     {Self-Remixing Loss}
21:     $\theta \leftarrow \theta - \eta\nabla_\theta\mathcal{L}$     {Update Student via SGD/Adam}
22:     $\theta_\mathcal{T} \leftarrow \alpha\theta_\mathcal{T} + (1-\alpha)\theta$     {EMA Teacher Distillation}
23: **end while**

---

*Code Listing 1.* PyTorch Implementation of SURF Iteration

```python
import torch

def train_surf_step(student, teacher, optimizer, mixtures,
                    K, alpha=0.999, mode='SR-FM'):
    """
    Implementation of Algorithm 1.
    mixtures: [B, 1, D] batch of observed signal mixtures.
    """
    B, _, D = mixtures.shape
    device = mixtures.device

    # --- 1. Teacher Estimates & Remixing ---
    with torch.no_grad():
        bar_X = teacher.separate(mixtures) # [B, K, D]

    # Global shuffle BK sources across the batch
    bar_X_flat = bar_X.view(B * K, D)
    perm = torch.randperm(B * K, device=device)
    tilde_X_1 = bar_X_flat[perm].view(B, K, D)

    # Synthetic mixtures [B, 1, D]
    tilde_M = tilde_X_1.sum(dim=1, keepdim=True)

    # --- 2. Flow Matching Path ---
    t = torch.rand(1, device=device)
```

```
    Z = torch.randn_like(tilde_X_1)

    # Consistency projection: mean(tilde_X_0) == mean(tilde_X_1)
    Z_centered = Z - Z.mean(dim=1, keepdim=True)
    tilde_X_0 = (tilde_M / K) + Z_centered

    # --- 3. PIT Alignment (Upsilon) ---
    with torch.no_grad():
        # Predict velocity at t=0 for optimal alignment
        v_0 = student(tilde_X_0, torch.zeros_like(t), tilde_M)
        tilde_X_1_aligned, upsilon_indices = optimal_pit(v_0, tilde_X_1, tilde_X_0)

    # Interpolant X_t and target velocity
    X_t = (1 - t) * tilde_X_0 + t * tilde_X_1_aligned
    target_drift = tilde_X_1_aligned - tilde_X_0

    v_pred = student(X_t, t, tilde_M)
    residual = v_pred - target_drift

    # --- 4. Loss & EMA Update ---
    if mode == 'RM-FM':
        loss = (residual**2).mean()
    else: # Self-Remixing
        # Assuming upsilon_indices has shape [B, K]
        upsilon_inv_indices = torch.argsort(upsilon_indices, dim=1)

        # Gather the residuals back to their unaligned positions
        residual = torch.gather(
            residual,
            1,
            upsilon_inv_indices.unsqueeze(-1).expand(-1, -1, D)
        )
        inv_perm = torch.argsort(perm)
        # Re-shuffling residual to match original mixture M
        res_flat = residual.view(B * K, D)[inv_perm]
        res_unshuffled = res_flat.view(B, K, D)
        loss = (res_unshuffled.sum(dim=1)**2).mean()

    optimizer.zero_grad()
    loss.backward()
    optimizer.step()

    # EMA Teacher Update
    with torch.no_grad():
        for s_p, t_p in zip(student.parameters(), teacher.parameters()):
            t_p.data.mul_(alpha).add_(s_p.data, alpha=1 - alpha)

    return loss.item()
```

### D.2. Audio Metrics

For completeness, we detail the metrics used in the empirical results for audio domain experiments.

**SI-SDR (Scale-Invariant Signal-to-Noise Ratio)**  SI-SDR (Le Roux et al., 2019) evaluates separation performance while addressing the amplitude scaling limitations of the traditional Signal-to-Distortion Ratio (SDR). It measures fidelity independent of gain by rescaling the target $x$ and estimate $\hat{x}$ such that the residual error is orthogonal to the target. The metric is defined as $\text{SI-SNR}(x, \hat{x}) = 10 \log_{10} \frac{||\alpha x||^2}{||\alpha x - \hat{x}||^2}$, where $\alpha = \frac{\hat{x}^\top x}{||x||^2}$ is a scalar coefficient that minimizes the residual energy.

**Single and Multi-source SI-SDR (FUSS)**  For FUSS (Wisdom et al., 2021), the result of separation of mixtures of 1 to 4 sources has to be evaluated. When the "mixture" is a single source, it is evaluated by the SI-SDR between the estimate

and the ground truth, i.e. SI-SNR$(\boldsymbol{x}, \hat{\boldsymbol{x}})$. This is labeled as $1S$ in Table 3. For mixtures $\boldsymbol{m}$ of $K \geq 2$ sources, the SI-SDR improvement (SI-SDRi), i.e., SI-SDRi$(\boldsymbol{m}, \boldsymbol{x}, \hat{\boldsymbol{x}})$ = SI-SDR$(\boldsymbol{x}, \hat{\boldsymbol{x}})$ − SI-SDR$(\boldsymbol{x}, \boldsymbol{m})$. This is called multi source SI-SDRi and labeled $K$Si, $K = 2, 3, 4$ in Table 3. Furthermore, the formulation of the SI-SDR is made robust with a formulation based on cosine similarity to maintain stability in cases of under-separation where estimates may be near zero. See Wisdom et al. (2021) for a detailed description.

**Under-separated, Equally-separated, and Over-separated Sources**   These classifications are detailed in Wisdom et al. (2021) and quantify the model's ability to detect the correct number of sources. Estimates are aligned to reference signals, disregarding effectively silent outputs. Performance is categorized by comparing the count of non-zero estimates ($M_{est}$) to active references ($M_{ref}$): *Under-separation* ($M_{est} < M_{ref}$) indicates missed sources; *Equal-separation* ($M_{est} = M_{ref}$) indicates a correct count; and *Over-separation* ($M_{est} > M_{ref}$) implies false positives or split sources.

**ESTOI (Extended Short-Time Objective Intelligibility)**   ESTOI (Jensen & Taal, 2016) is an algorithm that predicts speech intelligibility in environments with fluctuating noise, such as competing speakers. Unlike metrics that treat frequency bands independently, ESTOI accounts for spectral correlations across time segments (approx. 384 ms). By analyzing row- and column-normalized short-time spectrograms, it effectively captures spectro-temporal structures critical for intelligibility in modulated noise conditions.

**PESQ (Perceptual Evaluation of Speech Quality)**   Standardized as ITU-T Recommendation P.862, PESQ (Rix et al., 2001) objectively predicts the subjective quality of speech. The algorithm first aligns the degraded and reference signals in time and level, then transforms them into a loudness representation using a psychoacoustic model. A Mean Opinion Score (MOS) is calculated based on the aggregated disturbance between these representations, accounting for auditory masking effects.

**DNSMOS (Deep Noise Suppression Mean Opinion Score)**   DNSMOS P.835 (Reddy et al., 2022) is a non-intrusive, reference-free metric based on a deep neural network, designed to evaluate noise suppression when a clean ground truth is unavailable. Using a CNN trained on human ratings, it takes log power spectrograms as input to predict three subjective scores: speech quality (SIG), background noise quality (BAK), and overall quality (OVRL); in this work we report overall quality (OVRL) for this metric.

### D.3. Hybrid Teacher Formulation

To improve stability during the dynamic EMA update phase ($\alpha < 1$), the SURF model samples from a combined velocity field that switches between the initial frozen MixIT teacher baseline and the current parameterized velocity field $\boldsymbol{v}_{\theta_{\mathcal{T}}}$ based on a time threshold $\beta \in [0, 1]$. The hybrid velocity is defined as:

$$v_\beta(\boldsymbol{x}_t, t, \boldsymbol{m}) = \begin{cases} f_{\text{MixIT}}(\boldsymbol{m}) - \boldsymbol{x}_0 & t < \beta \\ \boldsymbol{v}_{\theta_{\mathcal{T}}}(\boldsymbol{x}_t, t, \boldsymbol{m}) & t \geq \beta, \end{cases}$$

where $\boldsymbol{x}_0 \in \mathbb{R}^{K \times d}$ is the initial source-array/noise state, $f_{\text{MixIT}}$ is the frozen teacher model, and $\boldsymbol{v}_{\theta_{\mathcal{T}}}$ is the dynamically updated teacher model.

Intuitively, this *hybrid teacher* trades off between the evolving flow model and the stable baseline. When $t < \beta$, the velocity field points directly from the source $\boldsymbol{x}_0$ to the MixIT estimate $f_{\text{MixIT}}(\boldsymbol{m})$. This acts as an "oracle" trajectory for a conditional flow sampling from a Dirac delta centered on the MixIT prediction. When $t \geq \beta$, the sampler switches to the standard EMA-updated teacher flow $\boldsymbol{v}_{\theta_{\mathcal{T}}}$.

We use the same $\beta$-schedule across all experiments. $\beta$ decays linearly from 1.0 to 0.0 over the first 200k iterations, gradually transitioning the training target from the fixed MixIT anchor to the fully dynamic SURF teacher. $\beta$ remains at 0.0 for all subsequent iterations. We maintain consistent architectures across regression-based and SURF-based unsupervised algorithms. For image separation, we use the NCSN architecture for both the frozen and updated teacher models. For audio separation, the frozen teacher is a ConvTasNet (Luo & Mesgarani, 2019), while the updated teacher utilizes an MB-TFLocoformer (Scheibler et al., 2025). Further experimental details for each domain can be found in the sections below.

## D.4. Image Separation

In the image separation setting, we implement both flow matching and separation models with a conditional NCSN architecture (Song et al., 2021) with 30M total parameters, where the regression model has the time conditioning and noise inputs deactivated. Flow models are sampled using a five-step Euler ODE solver, following Scheibler et al. (2025). For supervised regression, we directly train the model to predict the target output sources, given their average mixture. In flow matching models, we follow the mathematical framework in Section 2.3 to train a source separation model to learn the conditional flow matching velocity field of the underlying sources of a mixture, given the mixture itself. For all models and settings, we train with a squared error.

In the unsupervised setting, we proceed by first training the initial frozen MixIT teacher model. Here, we follow the guidelines in (Wisdom et al., 2020) and sample mixtures formed from single- and two-sample sources, with equal probability. This results in mixtures of mixtures that have two, three, and four sources with probabilities 25%, 50%, and 25%, respectively. For our proposed **SURF** method, the teacher model is then used to initialize the initial student and teacher models via Algorithm 1. For standard regression-based unsupervised separation, the teacher model is initialized with standard ReMixIT (Tzinis et al., 2022) or Self-Remixing (Saijo & Ogawa, 2023) algorithms. Full hyperparameters are provided in Table 6.

## D.5. Audio Separation

Following (Scheibler et al., 2025), we implement the Supervised Flow, ReMixIT, Self-Remixing, and both variants of our proposed method, SURF (ReMixIT) and SURF (Self-Remixing) with the MB-TFLocoformer architecture, consisting of a permutation-equivariant encoder-decoder network that processes a four-dimensional data tensor (time, band, source, feature) using a dual-path transformer backbone. The model employs a learnable Mel-band split front-end and alternates between Band-Source Joint Attention (BSJA) and Time-Source Parallel Attention (TSPA) blocks, which utilize Convolutional Multi-Head Self-Attention (CMHSA) to efficiently manage source interactions while conditioning on the mixture as an auxiliary token. Full hyperparameters for Libri2Mix and Audioset models are described in Tables 7 and 8, respectively. Following Scheibler et al. (2025), we train with a log-loss on the flow matching velocity squared error, and sample flow models using a five-step Euler ODE solver.

For the unsupervised regression models, Regression (Self-Remixing) and Regression (ReMixIT), the MB-TFLocoformer architecture, has only a single input channel (the single channel mixture conditioning), rather than mixture, noisy iterate, and time conditioning inputs. Following the flow-matching variants, we use the hyperparameters for Libri2Mix and AudioSet experiments (Tables 7 and 8). We train with a mean squared error loss.

For the frozen teacher model, we opted for a smaller ConvTasNet (Luo & Mesgarani, 2019) architecture with real-valued masks, following Wisdom et al. (2020). While in the image domain, we adhered to the formula in Wisdom et al. (2020) with training on mixtures of mixtures containing both single and two-sample mixtures, here we only train with the more difficult setting of mixtures of mixtures containing two-sample sources. This deviation is intended to demonstrate the validity of the unsupervised framework in the absence of any single source targets.

# E. Additional Results

## E.1. Further Ablations

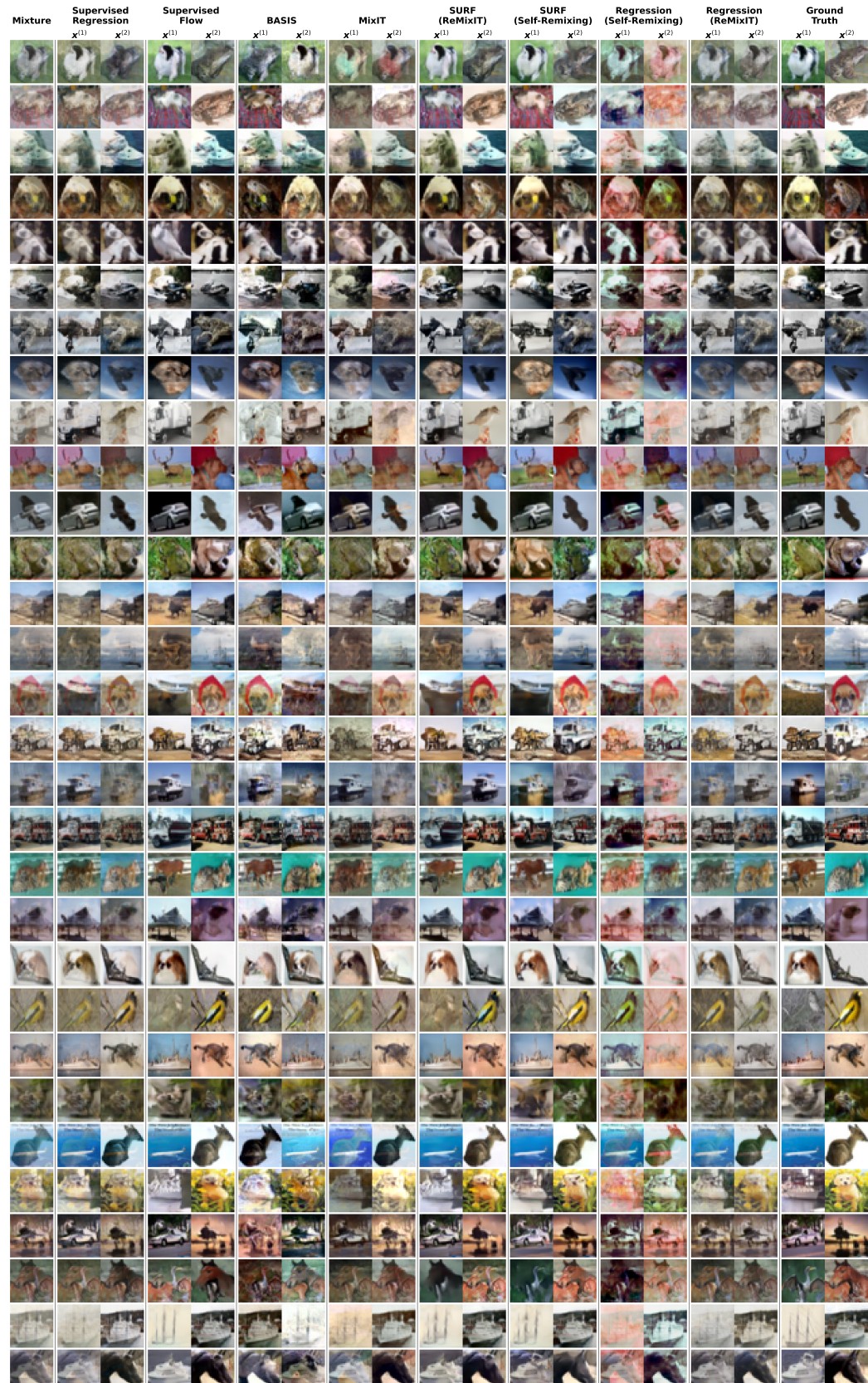

*Figure 3.* Qualitative examples for image separation on the CIFAR10 dataset.

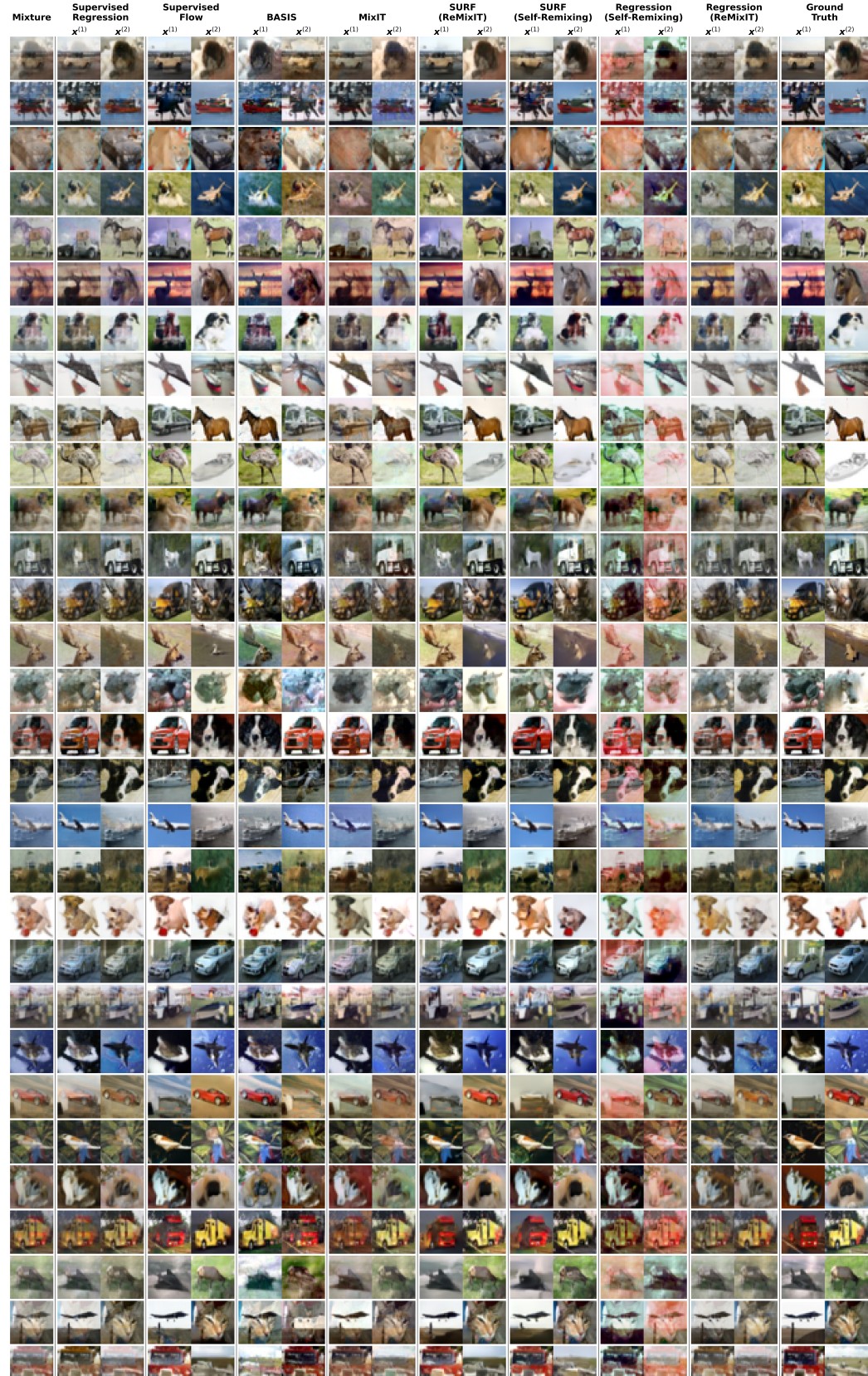

*Figure 4.* Further qualitative examples for image separation on the CIFAR10 dataset.

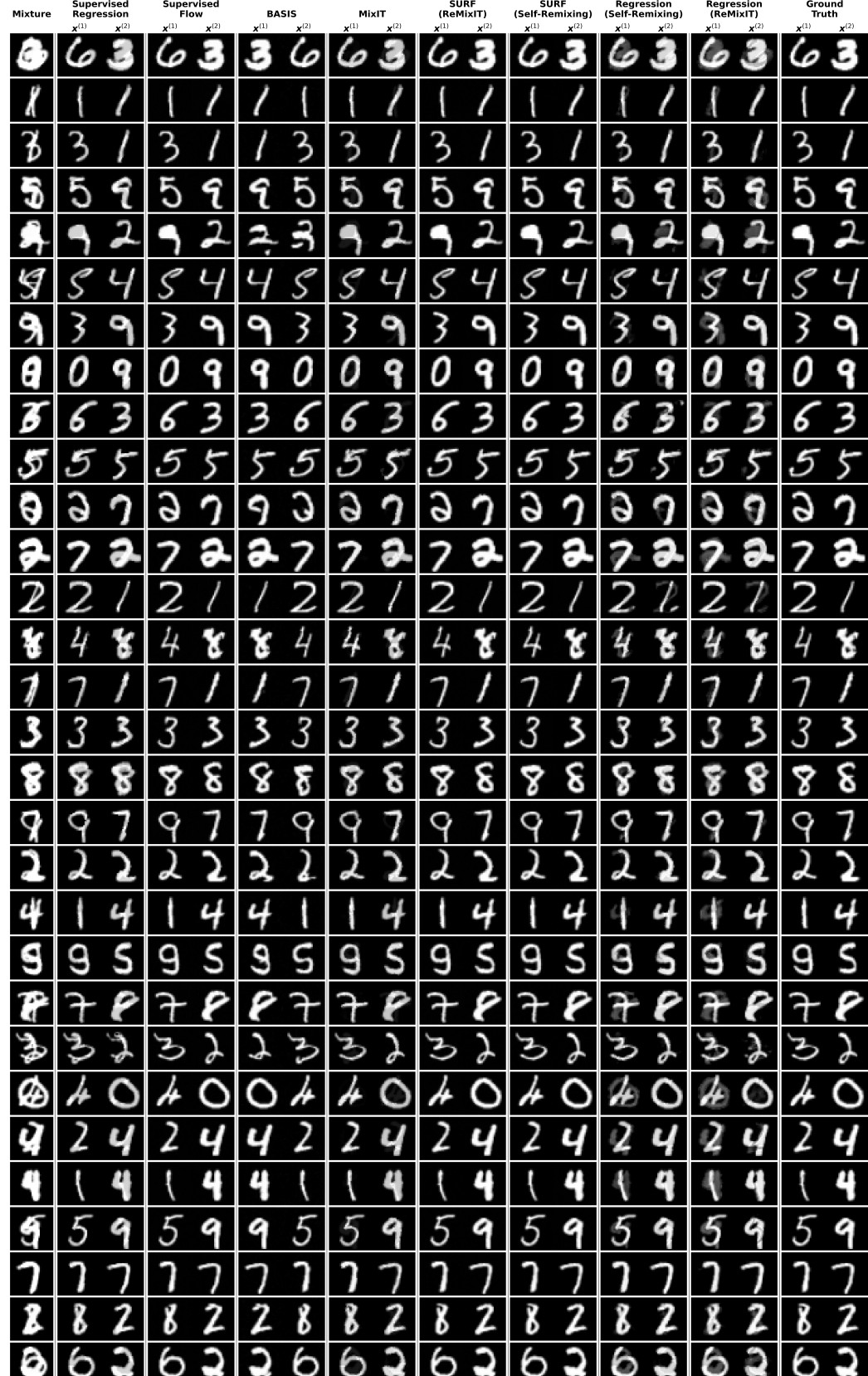

*Figure 5.* Qualitative examples for image separation on the MNIST dataset.

*Figure 6.* Further qualitative examples for image separation on the MNIST dataset.

Table 5. Summary of Notation

| Symbol | Dimensions | Description |
|---|---|---|
| *Indices & Constants* | | |
| $B$ | Scalar | Batch size (number of mixtures) |
| $K$ | Scalar | Number of sources per mixture |
| $d$ | Scalar | Signal dimension (e.g., time samples or flattened spectrogram) |
| $b, k$ | Indices | Mixture index $b \in \{1, \ldots, B\}$, Source index $k \in \{1, \ldots, K\}$ |
| $t$ | Scalar | Flow matching time step $t \in [0, 1]$ |
| $\mathbb{1}$ | $K \times 1$ | All-one column vector over sources |
| $\boldsymbol{P}^{\perp}$ | $K \times K$ | Projection onto the mixture-preserving orthogonal complement, $\boldsymbol{I}_K - \mathbb{1}\mathbb{1}^{\top}/K$ |
| $e_1$ | $K \times 1$ | First standard basis vector over source rows |
| *Original Data & Teacher* | | |
| $\boldsymbol{m}_b$ | $1 \times d$ | Single observed mixture $b$ |
| $\boldsymbol{M}$ | $B \times d$ | Batch of observed mixtures |
| $\bar{\boldsymbol{x}}_b$ | $K \times d$ | Teacher's source estimates for mixture $b$ |
| $\bar{\boldsymbol{X}}$ | $BK \times d$ | Batch of Teacher estimates (stacked $\bar{\boldsymbol{x}}_b$) |
| *Unsupervised Remixing (Finite Batch)* | | |
| $\Pi$ | $BK \times BK$ | Global permutation matrix that randomly shuffles sources across batch |
| $\widehat{\tilde{\boldsymbol{X}}}$ | $BK \times d$ | Batch of remixed sources (synthetic targets), $\widehat{\tilde{\boldsymbol{X}}} = \Pi\bar{\boldsymbol{X}}$ |
| $\tilde{\boldsymbol{m}}_b$ | $1 \times d$ | Synthetic mixture $b$ created by summing rows of $\tilde{\boldsymbol{x}}_b$ |
| $\tilde{\boldsymbol{M}}$ | $B \times d$ | Batch of synthetic mixtures |
| *Population Analysis (Limit $B \to \infty$)* | | |
| $\tilde{\boldsymbol{x}}_k$ | $K \times d$ | Pseudo-source matrix anchored to source $k$ |
| $\bar{\boldsymbol{x}}_{\text{bg}}^{(k,i)}$ | $1 \times d$ | Background teacher-estimated source drawn from $\bar{p}_{\theta_{\mathcal{T}}}$ |
| $\tilde{\boldsymbol{m}}_k$ | $1 \times d$ | Synthetic mixture for anchor $k$ ($\mathbb{1}^{\top}\tilde{\boldsymbol{x}}_k$) |
| $\tilde{\boldsymbol{x}}_{k,t}$ | $K \times d$ | Noisy flow state of the anchored pseudo-source matrix at time $t$ |
| *Permutation Invariant Alignment* | | |
| $\sigma_b$ | $K \times K$ | Optimal PIT permutation matrix for mixture $b$ |
| $\sigma_k$ | $K \times K$ | Optimal PIT permutation matrix for anchored pseudo-example $k$ |
| $\boldsymbol{\Upsilon}$ | $BK \times BK$ | Block-diagonal batch alignment, blkdiag($\sigma_1, \ldots, \sigma_B$) |
| *Student Flow Matching* | | |
| $\widehat{\tilde{\boldsymbol{X}}}_t$ | $BK \times d$ | Noisy flow state of the full batch at time $t$ |
| $\tilde{\boldsymbol{x}}_{b,t}$ | $K \times d$ | Noisy flow state of $b$-th batch of sources at time $t$ |
| $\boldsymbol{v}_{\theta}(\tilde{\boldsymbol{x}}_{b,t}, t, \tilde{\boldsymbol{m}}_b)$ / $\boldsymbol{v}_{\theta}(\widehat{\tilde{\boldsymbol{X}}}_t, t, \tilde{\boldsymbol{M}})$ | $K \times d$ / $BK \times d$ | Local / batch-level student velocity vector field |
| $\widehat{\boldsymbol{X}}$ | $BK \times d$ | Student's clean source estimates derived from flow |
| $\boldsymbol{R}$ | $BK \times d$ | Flow matching velocities minus targets |
| $\boldsymbol{Z}$ | $BK \times d$ | Batch Gaussian noise with vec($\boldsymbol{Z}$) $\sim \mathcal{N}(0, \boldsymbol{I}_{BKd})$ |

*Table 6.* NCSN Image Separation Backbone (30M total parameters)

| Hyperparameter | Value |
| --- | --- |
| Channel Multipliers | (1, 2, 2, 2) |
| Number of Filters | 64 |
| Number of Residual Blocks | 4 |
| Attention Resolutions | (16, 8) |
| Dropout | 0.1 |
| Conditional | True |
| Convolution Size | 3 |
| Embedding Type | fourier |
| FIR Kernel | (1.0, 3.0, 3.0, 1.0) |
| Fourier Scale | 16 |
| Initialization Scale | 0.0 |
| Nonlinearity | swish |
| Normalization | GroupNorm |
| Number of Scales | 2000 |
| Progressive Combine | sum |
| Progressive Input | input_skip |
| Progressive Output | output_skip |
| Resample with Convolution | True |
| Scale by Sigma | True |
| Skip Rescale | True |
| Sigma Min | 0.05 |
| Sigma Max | 0.5 |

*Table 7.* Audio Separation Backbone for Libri2Mix (36M total parameters)

| Hyperparameter | Value |
| --- | --- |
| Sampling Rate | 16 kHz |
| STFT Frame Length | 20 ms |
| STFT Overlap | 10 ms |
| STFT Window | Energy normalized Hamming |
| Magnitude Compression Exp. | 0.33 |
| Mel-bands | 80 |
| Feature Embedding Dim. | 192 |
| BSJA Kernel Size (Time) | 5 |
| TSPA Kernel Size (Time, Freq) | (5, 3) |
| Number of Blocks | 6 |
| Number of Output Sources | 4 |

*Table 8.* Audio Separation Backbone for AudioSet (36M total parameters)

| Hyperparameter | Value |
| --- | --- |
| Sampling Rate | 16 kHz |
| STFT Frame Length | 20 ms |
| STFT Overlap | 10 ms |
| STFT Window | Energy normalized Hamming |
| Magnitude Compression Exp. | 0.33 |
| Mel-bands | 80 |
| Feature Embedding Dim. | 384 |
| BSJA Kernel Size (Time) | 5 |
| TSPA Kernel Size (Time, Freq) | (5, 3) |
| Number of Blocks | 6 |
| Number of Output Sources | 4 |

*Table 9.* Comprehensive ablation studies for SURF. We evaluate the impact of EMA decay ($\alpha$), batch size ($B$), and the hybrid teacher $\beta(t)$ schedule on CIFAR-10 image separation. Best results for each ablation are bolded.

| Ablation | Variant | SURF (Self-Remixing) | | | SURF (ReMixIT) | | |
|---|---|---|---|---|---|---|---|
| | | PSNR ↑ | SSIM ↑ | LPIPS ↓ | PSNR ↑ | SSIM ↑ | LPIPS ↓ |
| **EMA** $\alpha$ | $\alpha = 0$ | 16.55 | 0.631 | 0.077 | 16.63 | 0.635 | 0.075 |
| | $\alpha = 0.9$ | 18.55 | 0.733 | 0.054 | 18.62 | 0.739 | 0.051 |
| | $\alpha = 0.99$ | 19.01 | 0.741 | 0.041 | 18.99 | 0.741 | 0.043 |
| | $\alpha = 0.999$ | **19.49** | **0.751** | **0.037** | **19.73** | **0.756** | **0.036** |
| | $\alpha = 0.9999$ | 13.36 | 0.372 | 0.101 | 13.55 | 0.401 | 0.098 |
| **Batch Size** $B$ | $B = 1$ | 8.79 | 0.071 | 0.379 | 15.87 | 0.596 | 0.094 |
| | $B = 4$ | 16.46 | 0.629 | 0.078 | 16.88 | 0.657 | 0.075 |
| | $B = 16$ | 17.49 | 0.681 | 0.065 | 17.65 | 0.697 | 0.063 |
| | $B = 64$ | 18.11 | 0.730 | 0.056 | 18.85 | 0.712 | 0.048 |
| | $B = 256$ | **19.49** | **0.751** | **0.036** | **19.73** | **0.756** | **0.036** |
| **Teacher** $\beta(t)$ | $\beta = 0.1$ | 8.15 | 0.078 | 0.380 | 8.48 | 0.081 | 0.372 |
| | $\beta = 0.5$ | 18.46 | 0.721 | 0.051 | 18.41 | 0.719 | 0.053 |
| | $\beta = 1.0$ | 17.30 | 0.701 | 0.060 | 18.08 | 0.723 | 0.055 |
| | Proposed Sched. | **19.49** | **0.751** | **0.037** | **19.73** | **0.756** | **0.036** |

