# OpenReview forum: "SURF: Separation via Unsupervised Remixing Flow"
_ICML.cc/2026/Conference — ICML 2026 regular_

### Official Review · Reviewer_veC9 · 2026-03-11

**Soundness:** 3
**Presentation:** 3
**Significance:** 3
**Originality:** 2
**Overall Recommendation:** 5
**Confidence:** 2

**Summary:**

This paper presents SURF, a generative framework for single-channel source separation that learns directly from observed mixtures in unsupervised way. The authors bridge the gap between deterministic regression-based remixing techniques and generative flow matching by relating flow velocity to source estimates, allowing for a student model to be bootstrapped from a teacher model's estimates through a remixing step. The method incorporates two variants based on existing self-supervised techniques, ReMixIT and Self-Remixing. Empirical evaluations across image and audio benchmarks, including MNIST, CIFAR-10, Libri2Mix, and AudioSet, demonstrate that SURF achieves state-of-the-art results for unsupervised separation by significantly outperforming existing non-generative baselines in both reconstruction fidelity and perceptual quality.

**Compliance With Llm Reviewing Policy:**

Affirmed.

**Final Justification:**

I am increasing my score as the rebuttal’s benchmarks and complexity analysis resolved my concerns.

**Key Questions For Authors:**

* It is unclear whether the final evaluations utilize the Teacher or the Student model. While the framework updates the Teacher via EMA from the Student, the text does not explicitly state which weights were used for the reported benchmarks.
* In Table 3, several methods, particularly the MixIT baseline, show similar performance on the LibriSpeech + FUSS metrics. Providing standard deviations and a supervised benchmark for this specific test would help clarify if the task is already near saturation and if SURF's advantages are statistically significant.

**Limitations:**

yes

**Strengths And Weaknesses:**

Strengths:

* The paper introduces a novel integration of flow matching models into the source separation task by bridging the structural discrepancy between deterministic remixing algorithms and the velocity fields used in flow matching.
* The proposed SURF method establishes a new state-of-the-art for unsupervised source separation, significantly outperforming existing unsupervised regression-based techniques like MixIT, ReMixIT, and Self-Remixing.
* By leveraging a generative framework, the model achieves superior perceptual quality and distributional resemblance compared to traditional regression models, which often suffer from unrealistic artifacts.


Weaknesses:

* The paper lacks a comprehensive time complexity analysis regarding inference, which is particularly relevant since generative flow-based models typically require multiple ODE solver steps, potentially making them significantly slower than the single-pass "old" regression-based methods they are compared against.
* There is a lack of complexity analysis regarding the transition of the teacher model from a regression model to a Flow Matching (FM) model. Specifically, the training process requires the student to be trained from teacher estimates, which in the FM framework involves ODE sampling that can significantly increase the computational cost and time during the training iterations.
* The presentation of the paper would be improved if Algorithm 1 included more detailed comments or annotations to clarify the multi-step remixing and flow-matching process for the reader.

---

> ### Author Rebuttal · Authors · 2026-03-31
>
> Thank you for your constructive feedback and for noting that SURF represents a "new state-of-the-art framework" that mitigates the unrealistic artifacts of traditional regression models. We appreciate the opportunity to clarify the complexity and implementation details of our approach. [*Anonymized rebuttal page*](https://separation-unsupervised-remixing-flow.github.io/icml2026-rebuttal/).
>
> **Time Complexity and Inference Latency**
> We acknowledge that generative FM models involve more compute than single-pass regression. However, SURF is designed for efficiency: our flow-based models (approx. 50M parameters for audio) require only 5 NFE (Number of Function Evaluations) during sampling. Using a 5-step Euler solver with a nonlinear schedule achieves high fidelity while remaining computationally competitive with traditional methods. Ultimately, specific latency is hardware-dependent, but the 5-step requirement ensures that "real-time" performance is within reach for modern accelerators.
>
> **Computational Cost of Training and Sampling Steps**
> The transition from a regression teacher to an FM student is more efficient than it may appear. During training, we found that sampling with even fewer steps is sufficient; we use $NFE=2$ for the teacher sampling step. This results in a minimal 15% overhead for the overall training iteration time compared to supervised FM baselines. In wall-clock time on audio tasks with Cloud TPU v4 accelerators, this is ~2.2s per training iteration for FLOSS and SURF with a predictive (frozen) teacher, and ~2.5s per training iteration for SURF with a flow-based teacher. On image tasks, this is 0.12s per iteration and 0.15s per iteration, respectively.
>
> It is important to note the architectural configurations used: while MixIT employs its original ConvTasNet backbone, all other evaluated models (SURF and the regression-based ReMixIT/Self-Remixing baselines) share a common MB-TFLocoformer architecture. These models are identical in terms of layers, width, parameters, and STFT hyperparameters, ensuring a fair comparison within that group.
>
> **Algorithm 1 Annotations and Clarity**
> We agree that the multi-step remixing and FM process is complex. Due to strict double-column space constraints in the main text, we were forced to be brief. However, we will include a significantly expanded, heavily annotated version of Algorithm 1 in the Appendix, as well as PyTorch-style pseudocode to assist researchers in implementation.
>
> Overall, SURF uses a three-phase loop: 1) *Generative Remixing*: an EMA teacher estimates sources from mixtures, shuffling them batch-wide to create synthetic mixtures. 2) *Flow Matching*: a conditional path is defined using PIT-aligned interpolants between projected noise and estimates. 3) *Optimization*: the student is updated via ReMixIT (RM-FM) or Self-Remixing (SR-FM) drift residuals, followed by an EMA update of the teacher. This structure enables unsupervised separation by matching the velocity of the teacher-guided generative remixing process.
>
> **Teacher vs. Student Weights in Benchmarks**
> We apologize for the ambiguity. In alignment with standard practice in generative modeling (e.g., flow matching and diffusion), all reported benchmarks utilize the Exponential Moving Average (EMA) of the model weights. In our SURF framework, these EMA weights are what define the Teacher model. We will update the manuscript to explicitly state that the final evaluations are conducted using the Teacher (EMA) weights.
>
> **Supervised baseline and statistical significance.** We thank the reviewer for suggesting stronger statistical grounding. To address saturation concerns in Table 3, we added a *supervised baseline* and *standard error (SEM)* for all metrics. The supervised baseline ($18.21\pm0.27$ dB SI-SNR; $2.58\pm0.02$ DNSMOS) significantly outperforms our best unsupervised model (SURF Self-Remixing: $15.23\pm0.26$ dB; $2.57\pm0.01$). Updated results (from a later checkpoint) show the task is not saturated; substantial room for improvement remains. SURF achieves *significant gains* over MixIT in SI-SNR, PESQ, and DNSMOS; 95% CIs do not overlap ($14.18\pm0.20$ dB, $2.79\pm0.02$, $2.53\pm0.01$). At $N=1000$, these represent a robust step toward the supervised upper bound. We updated the [*rebuttal page*](https://separation-unsupervised-remixing-flow.github.io/icml2026-rebuttal/) and manuscript with these SEMs and benchmarks.

---

> > ### Author Rebuttal · Reviewer_veC9 · 2026-04-03
> >
> > I thank the authors for their thorough response, as the additional complexity analysis and supervised benchmarks effectively resolve my concerns regarding the model's efficiency and statistical significance.

---

### Official Review · Reviewer_Gtkq · 2026-03-12

**Soundness:** 3
**Presentation:** 3
**Significance:** 4
**Originality:** 4
**Overall Recommendation:** 5
**Confidence:** 4

**Summary:**

This paper proposes SURF, a framework for unsupervised single-channel source separation that cleverly combines self-supervised remixing techniques (specifically ReMixIT and Self-Remixing) with conditional flow matching (FM). The core mechanism involves bootstrapping a student flow model using a teacher model initially trained with MixIT. By constructing synthetic mixtures from the teacher's estimates, the student is trained using FM-style losses derived for both the ReMixIT and Self-Remixing paradigms. The authors back this up with solid theoretical work, including a population analysis that links their proposed losses back to supervised FM, and an interesting Wake-Sleep interpretation of the process. Empirically, the method performs very well on both image (MNIST, CIFAR-10) and audio (Libri2Mix, AudioSet) benchmarks, setting a new state-of-the-art for unsupervised methods in these contexts.

**Compliance With Llm Reviewing Policy:**

Affirmed.

**Key Questions For Authors:**

1. **Teacher Sensitivity:** How sensitive is SURF to the quality of the initial teacher? What happens to performance if you initialize with a much weaker MixIT model, or if you ablate the EMA update entirely? Does SR-FM degrade gracefully?
2. **Computational Costs:** Could you provide details on the computational costs (training time, memory usage, inference latency) compared to the regression baselines and supervised FM? How many ODE steps are used during inference, and what is the trade-off between quality and step count?
3. **Missing Ablations:** Can you provide ablations on the time weighting $\lambda(t)$, the handling of the $(1-t)$ factors, and the use of PIT vs. no-PIT within the FM framework? How much of the gain is specifically driven by the FLOSS alignment?
4. **Source Count ($K$) Robustness:** How does SURF behave under source-count mismatch (e.g., training with $K=2$ but testing on 3-4 sources, and vice versa)? Can this approach be extended to variable-$K$ separation with a principled stopping criterion? Regarding the performance drop on AudioSet for 3-4 sources, is this due to remixing bias or capacity limits?
5. **Unsupervised Baselines:** Could you compare SURF (even on a subset of tasks) to unsupervised diffusion posterior sampling methods that utilize clean priors? This would help position SURF more clearly in scenarios where clean priors might actually be available.

**Limitations:**

No. While the authors discuss the method generally, they need to expand on the potential risks of hallucination and bias amplification inherent in generative models used for separation. A discussion on how to implement safeguards (like uncertainty estimates or consistency checks) for downstream deployments is necessary.

**Strengths And Weaknesses:**

**Strengths**

1. **Technical Novelty:** The paper introduces a principled way to bridge self-supervised remixing (which is traditionally regression-based) with flow matching. Relating the FM velocities to regression-style clean estimates via a Tweedie-like relation is a neat theoretical contribution.
2. **Smart Adaptations:** Adapting PIT-style alignment to FM training paths (building off FLOSS) and proving the validity of permutation-conditioned paths through symmetry arguments is rigorous and well-executed.
3. **Strong Theoretical Grounding:** The population analysis is a highlight. It cleanly shows how the proposed RM-FM and SR-FM losses deviate from supervised FM, specifically isolating the dependence on teacher errors versus cross-source correlations. The Wake-Sleep framing also nicely clarifies the teacher-student dynamics.
4. **Comprehensive Experiments:** The evaluation spans both images and audio, utilizing a diverse set of metrics (including crucial perceptual and distributional measures like FID and DNSMOS). SURF shows consistent gains over strong unsupervised baselines and even approaches supervised FM performance on certain tasks. The inclusion of universal separation results (AudioSet to FUSS) provides a good look at over/under-separation behavior.
5. **Clarity and Significance:** The problem setup and notation are clear. The method addresses a highly practical limitation in the field—learning strong generative separators without needing clean source data. This could easily serve as a template for integrating remixing self-supervision with modern generative modeling.

**Weaknesses**

1. **Dependence on Teacher Quality:** The approach seems to rely on having a reasonably strong initial teacher (MixIT) and likely inherits its biases. The paper doesn't fully characterize the conditions under which the student FM can meaningfully surpass a weak teacher.
2. **Stability and Convergence:** The dynamics of the teacher-student EMA setup are only briefly touched upon. There is a lack of theoretical or empirical diagnostics regarding potential failure modes, such as confirmation bias during training.
3. **Missing Ablations:** The experimental section lacks some key ablations. For instance, there is no study on the impact of teacher quality, the EMA decay rate ($\alpha$), the batch size ($B$) used for remixing, or the specific impact of using PIT versus no-PIT during FM training. Furthermore, the sensitivity to the time weighting $\lambda(t)$ and choices around the Tweedie approximation aren't explored.
4. **Scalability and Mismatch:** In the universal (AudioSet) results, performance noticeably drops when handling 3-4 sources. The method's behavior under source-count mismatches (training on $K=2$ but testing on more) and its general scalability to larger $K$ are not analyzed.
5. **Incomplete Baselines:** The paper misses comparisons to recent unsupervised diffusion/separation methods that *do* use clean priors (but no clean mixtures), such as posterior sampling or hybrid solvers. While SURF operates in a different regime (no clean data at all), discussing and comparing against these methods would better contextualize the trade-offs of not having single-source corpora.
6. **Minor Presentation Issues:** There are some notation collisions (e.g., $\Gamma/\Upsilon$ vs. $\sigma$) and Algorithm 1 interleaves RM-FM and SR-FM without much discussion on when one might be preferred over the other. Inference-time details (ODE solver specifics, step counts, computational cost) are also missing.

---

> ### Author Rebuttal · Authors · 2026-03-31
>
> Thank you for your rigorous review and for recognizing the theoretical grounding of SURF. We are pleased you found our adaptation of PIT-style alignment to Flow Matching (FM) both "rigorous and well-executed." Below, we address your concerns with additional data and clarification. [_Anonymized rebuttal page_](https://separation-unsupervised-remixing-flow.github.io/icml2026-rebuttal/).
>
> **Reliance on Initial Teacher Quality and Inherited Biases**
> Initializing from a weak teacher is a frontier in unsupervised separation. However, prior research on Self-Remixing (Saijo et al., 2023) demonstrated that convergence is achievable even from a random initialization. We anticipate SURF will exhibit similar robustness and will incorporate a discussion of "Remixing-based Unsupervised Source Separation from Scratch" (arXiv:2309.00376), as well as experiments for the flow-based setting, in the final version of this paper.
>
> **Teacher-Student EMA Dynamics and Failure Modes**
> The dynamics of the teacher-student EMA framework were characterized by Saijo et al. (2023), establishing that omitting EMA updates causes models to inherit teacher artifacts, leading to confirmation bias and overfitting. Our ablation study on the EMA decay parameter finds that performance is stable across a wide range of values. We will include these details regarding the influence of EMA on training dynamics in the manuscript.
>
> **Ablation Studies: EMA, Batch Size, $\lambda(t)$, and PIT**
> We note existing ablation studies on PIT and $\lambda(t)$ in Scheibler et al. (2025), which demonstrated that PIT losses provide a +1.75dB improvement over non-PIT variants. To address hyperparameter sensitivity, we performed [_additional studies on the CIFAR10 dataset_](https://separation-unsupervised-remixing-flow.github.io/icml2026-rebuttal/):
>
> * **Hybrid-teacher ($\beta$):** Compared fixed values $\beta \in \{0, 0.1, 0.5, 1.0\}$ against our proposed $\beta(t)$ schedule; the schedule provided the best balance between early stability and late refinement.
> * **EMA Rate ($\alpha$):** Tested $\alpha \in \{0, 0.9, 0.99, 0.999, 0.9999\}$; the model was robust between 0.99 and 0.9999.
> * **Batch Size ($B$):** Evaluated $B \in \{1, 4, 16, 64, 256\}$; performance plateaus after $B=64$ as permutation diversity saturates.
>
> **Tweedie Approximation Alternatives**
> We investigated an alternative estimator for clean source prediction in Self-Remixing. We proved this alternative is equivalent to the proposed estimator up to a time-dependent re-weighting of the FM objective (Eq. 9 in Lipman et al.). We retained the standard parameterization for clarity but detailed the alternative derivation in Appendix C.3.
>
> **Performance Drops for >2 Sources**
> This drop is primarily due to a domain mismatch between training (AudioSet) and evaluation (FUSS). AudioSet examples contain ~2.7 sources on average. MixIT baselines are trained on "mixtures-of-mixtures" (up to 8 sources, ~5.4 average), explaining their advantage in high-count scenarios. However, SURF significantly outperforms MixIT on 1- and 2-source examples where the distribution is more aligned.
>
> **Comparisons to Unsupervised Methods with Clean Priors**
> We compare SURF to BASIS (Tables 1 and 2), a posterior sampling method using a pretrained diffusion prior. Despite SURF having no access to clean source data, it achieves comparable or superior FID/IS scores. Methods using clean priors are not strictly "unsupervised" from scratch; SURF addresses "in-the-wild" scenarios where clean examples are unavailable.
>
> **Notation and Algorithm 1 Guidance**
> We will simplify $\Upsilon$ to $\Gamma^*$ to streamline notation. Regarding variant selection:
> * **ReMixIT:** Effective in high-fidelity domains but prone to confirmation bias.
> * **Self-Remixing:** More robust to domain mismatches and weak teachers due to its sum-based objective.
>
> **Computational Costs and ODE Details**
> SURF uses a 5-step Euler solver with a nonlinear schedule (Appendix D.3). With architectural parity (MB-TFLocoformer), the base cost per iteration is identical to regression. Sampling overhead is minimal: 85% of iteration time is spent on the forward/backward pass, with only 15% for the $NFE=2$ teacher sampling step.
>
> **Hallucination Risks and Safeguards**
> Our model learns $p(x|y)$ directly rather than a generative prior $p(x)$, reducing potential bias. The FLOSS backbone provides a structural safeguard: it is "mixture-consistent," ensuring separated sources sum exactly to the input mixture ($\sum_{i=1}^K x_t^{(i)} = m$), preventing the generation of unrelated hallucinated audio.

---

### Official Review · Reviewer_3AeJ · 2026-03-13

**Soundness:** 3
**Presentation:** 1
**Significance:** 3
**Originality:** 2
**Overall Recommendation:** 4
**Confidence:** 3

**Summary:**

The paper

- proposes SURF, a self-/unsupervised source separation method based on conditional flow matching
- combines supervised flow-matching ideas with self-supervised remixing: a teacher separates synthetically mixed signals to create pseudo-paired data for a student
- provides analysis/insight into the optimized self-supervised objective (including behavior as the number of mixtures $B$ grows)
- validates the proposed on both image and audio separation tasks

**Compliance With Llm Reviewing Policy:**

Affirmed.

**Final Justification:**

I thank the authors for their rebuttal. My concerns have been addressed, and I am therefore raising my score.

**Key Questions For Authors:**

- SURF (ReMixIT loss) and SURF (Self-Remixing) perform very similarly: can you explain when/why the two objectives differ in practice and provide recommendations on when to use each?
- Table 3: why does performance degrade so sharply for SURF with respect to metrics 1S, 2Si, 3Si, 4Si, in particular it drops from ~26 to ~ 8; while for MixIT this behavior is less pronounced, the metrics drop from ~11 to ~9.5?

**Limitations:**

The method
- relies on the teacher quality and stability; failure modes when the teacher is weak or biased are unclear
- may be sensitive to remixing design choices and hyperparameters (e.g., $\beta$, EMA rate, batch size $B$)

**Strengths And Weaknesses:**

### Strengths

- Presents an unsupervised conditional flow-matching approach for source separation.
- Extension of the supervised flow-matching separation framework of Scheibler et al., 2025 to self-supervised training setting
- Theoretical discussion for the self-supervised objective, namely in the asymptotic regime of the number of considering mixtures of mixtures

### Weaknesses

Writing / notation
- The paper is hard to follow due to heavy, overloaded notation (Secs. 3.2 and 3.3), making equations difficult to verify
- Sec. 3.3 largely re-derives the previous section for batched mixtures
- The appendix notation table helps, but the main text could be simplified.

Method/implementation and validation
- teacher initialization is underspecified: how is a MixIT-style model adapted to initialize a FLOSS/flow-based teacher given the conceptual mismatch (the former is flow matching (time constrained for instance) whereas the other is not)?
- The hybrid-teacher parameter $\beta$, App. D.2, lacks ablations (sensitivity, initialization choices) and details on how it is set in practice
- analysis of training-time and comparisons between methods is not provided; namely this matters for the considered teacher-student/EMA setting
- the choice of the batch of mixtures is not ablated as well

minor issues
- Col. 2, line 103: missing transpose: $ 1 m \rightarrow 1 m^\top$.
- Col. 2, line 104: the distribution of $z$ should be Gaussian with covariance $I_{K \times d}$
- Revisit the usage of `\citet` vs `\citep` as their are Multiple inconsistencies (see for instance Col. 1, lines 120/230/373)

Missing references
- source separation can also be tackled from training-free (inference-time) perspective, where one cast source separation as an inverse problem and uses a pre-trained diffusion/flow model as prior to guide the generation; see Daras et al. 2024 [1] for a review, in particular, training-free source separation was tackled in the work by Yazid et al. 2025 [2]


---

.. [1] Daras, Giannis, et al. "A survey on diffusion models for inverse problems." (2024).

.. [2] Janati, Yazid, et al. "A mixture-based framework for guiding diffusion models." (2025).

---

> ### Author Rebuttal · Authors · 2026-03-31
>
> We thank the reviewer for their thoughtful consideration of our work. We are encouraged that the reviewer highlighted our extension of flow matching to the unsupervised setting, as well as the theoretical discussions. Below, we address their concerns point-by-point.
>
> **Notation Complexity (Sections 3.2, 3.3)**
> We agree that the indexing required to track batch items, student/teacher estimates, and PIT permutations can be hard to follow. To improve readability in the revised manuscript, we will streamline notation by adopting vector-form notation in Sections 3.2 and 3.3. This allows us to consolidate Equations 16–24 by omitting explicit batch indices where the context of i.i.d. operations are clear. This will make the underlying flow-matching logic more accessible without sacrificing mathematical rigor.
>
> **Distinction Between ReMixIT (3.2) and Self-Remixing (3.3)**
> We appreciate the opportunity to clarify the algorithmic differences between Section 3.2 and Section 3.3. While both are unsupervised source separation methods, they utilize fundamentally different training objectives:
>
> * **ReMixIT:** Directly minimizes the error between student estimates and teacher source predictions. It is highly effective when the teacher is stable and reliable.
> * **Self-Remixing:** Minimizes a modified target involving the sums of estimates. This "weaker" supervision is critical when there is a domain mismatch (e.g., training on AudioSet but evaluating on FUSS) or when the teacher has not yet been refined via EMA. It prevents the student from prematurely over-fitting to biased teacher outputs.
>
> We will revise Section 3.3 to explicitly contrast these two paths to help practitioners choose the appropriate objective for their specific data constraints.
>
> **Teacher Initialization and EMA**
> The student model is initialized with random weights and initially supervised by a frozen teacher. As training progresses, we transition to an EMA-updated teacher using a $\beta$-interpolation schedule (detailed in Section 5 and Appendix D.2-4). This prevents the "feedback loop" instability often seen in self-supervised learning. Regarding teacher quality, prior work in Self-Remixing (arXiv:2309.00376) demonstrates that even starting from a randomly initialized teacher can lead to convergence; we will incorporate a discussion of these stability guarantees in the revised manuscript.
>
> **Sensitivity Analysis (Ablation Studies)**
> To address concerns regarding hyperparameter sensitivity, we performed additional studies on CIFAR10: [**link**](https://separation-unsupervised-remixing-flow.github.io/icml2026-rebuttal/).
>
> * **Hybrid-teacher ($\beta$):** We compared fixed values $\beta \in \{0, 0.1, 0.5, 1.0\}$ against our proposed $\beta(t)$ schedule. The schedule provided the best balance between early-stage stability and late-stage refinement.
> * **EMA Rate ($\alpha$):** We tested $\alpha \in \{0, 0.9, 0.99, 0.999, 0.9999\}$. We found the model robust to rates for most values, confirming that our chosen parameters are not "brittle."
> * **Batch Size ($B$):** We evaluated $B \in \{1, 4, 16, 64, 256\}$. Performance gain plateaus after $B=64$ because the diversity of permutations within the batch saturates, ensuring the PIT loss has sufficient signal.
>
> **Computational Efficiency and Training Overhead**
> Our framework maintains high efficiency due to the fact that we found that using a low number of sampling steps ($NFE=2$) was sufficient for the teacher.
>
> * There is only a 15% overhead for teacher sampling. The majority of the wall-clock time (85%) remains the forward / backward pass of the student model.
> * **Audio Tasks:** Both FLOSS and SURF models with a frozen teacher take ~2.2s per training step on Cloud TPU v4 accelerators. During teacher-student SURF training with flow teacher sampling, this increases to ~2.5s.
> * **Image Tasks:** FLOSS and SURF with a frozen teacher take ~0.12s per training step, whereas SURF with teacher sampling training takes ~0.15s.
>
> **High Source-Count Performance (Table 3)**
> The performance drop on multi-source mixtures (3Si, 4Si) is primarily a result of training distribution mismatch. MixIT was trained on "mixtures of mixtures" (up to 8 sources), while SURF focuses on recovering the single-mixture distribution. Since SURF is architecturally flexible, the training mixture distribution can always be shifted to match higher source counts. We'll clarify this training distribution-dependent trade-off.
>
> **Citations and Technical Corrections**
> We thank the reviewer for pointing out these references. We will include Daras et al. (2024) (in addition to their 2025 work already cited) and Janati et al. (2025). We note a crucial distinction: Janati et al. (2025) relies on a pretrained diffusion prior for sources trained using isolated sources. In contrast, SURF remains strictly unsupervised, requiring only mixtures. We have corrected the typos on lines 103 ($1_m^\top$) and 104 ($I_{K \times d}$) and will standardize citation styles.

---

> > ### Author Rebuttal · Reviewer_3AeJ · 2026-04-01
> >
> > I appreciate the authors response.
> >
> > My main concerns have been addressed. I still did not fully understand the performance drop reported in Table 3.
> > I would appreciate further clarification.

---

> > > ### Author Response · Authors · 2026-04-02
> > >
> > > We thank the reviewer for their continued engagement and for the opportunity to further clarify the performance dynamics in Table 3. Ultimately, the observed performance drop on higher source counts stems from an artificial distribution shift during evaluation—a discrepancy that does not impact the model's strong capabilities on primary downstream tasks, such as single-source denoising.
> > >
> > > **Distribution Shift in Multi-Source Mixtures (>2 Sources in Table 3)**
> > > We acknowledge the observed performance degradation for mixtures with more than two sources, which we attribute to a distribution shift between training and evaluation. This discrepancy is directly explained by a *fundamental difference in remixing strategies:*
> > >
> > > * *MixIT:* The input to the model is always the sum of two distinct mixtures (a “mixture of mixtures”). In the case of AudioSet, this artificially doubles the training source count to an average of **~5.4 sources per input**. Consequently, evaluating on 4-source mixtures remains comfortably in-distribution for MixIT.
> > > * *SURF & Regression-based Remixing:* In contrast, the teacher-student scheme operates strictly on single AudioSet examples. This method does *not* artificially increase the source count, maintaining the true dataset average of **~2.7 sources per input**. Therefore, evaluating on 3- and 4-source mixtures forces these models into an out-of-distribution regime.
> > >
> > > Our empirical findings indicate that SURF variants are more robust to this domain shift than their regression-based counterparts, as evidenced by their improved performance on >2 source examples. Nonetheless, they are outperformed by the MixIT baseline, which does not experience a comparable test-time shift on 3- and 4-source mixtures. (Conversely, MixIT models exhibit a more pronounced performance drop on 1- and 2-source examples). We have incorporated this discussion into the final manuscript to provide complete context.
> > >
> > > Importantly, while the operating point of these models can be shifted by adjusting the source count in training mixtures, our results demonstrate that SURF models are significantly more faithful in capturing single- and two-source distributions—a primary objective of unsupervised sound separation. The superior performance on 1-source (1S) examples is not incidental; it indicates that SURF variants capture the underlying single- and two-source data distributions more effectively than MixIT or ReMixIT baselines, and they achieve this in a purely unsupervised manner.
> > >
> > > Edit Apr 7, 2026: We’d like to further address the original concern, about why there is an inherent drop in performance as the number of sources (N) increases, in particular going from N=1 to 2. Beyond distribution shift, this drop is an expected consequence of the task's fundamental characteristics. Assuming equal-power sources, the input SNR is $10 \log_{10}(1 / (N - 1))$. At N=1, the input is noise-free (requiring an identity mapping). For N=2, 3, 4, the SNR drops to 0, -3.0, and -4.7 dB. Therefore, an ideal algorithm will naturally show peak performance at N=1, a sharp decline at N=2, and smooth degradation thereafter. Conversely, MixIT struggles at N=1 because it is trained solely on mixture-of-mixtures and never learns the requisite single-source identity mapping.
> > >
> > > **Performance on 2-Source Mixtures (LibriSpeech + FUSS) and Statistical Significance**
> > > To further contextualize the model's capabilities on a strictly single-source reconstruction task (where the aforementioned distribution shift is mitigated), we have updated our evaluation on the noisy speech samples (LibriSpeech + FUSS). The updated table and detailed metrics can be viewed anonymously here: https://separation-unsupervised-remixing-flow.github.io/icml2026-rebuttal/
> > >
> > > *(Note: These updated metrics reflect the fully converged model, showing slight improvements over the intermediate checkpoint used in the original manuscript).*
> > >
> > > In this updated evaluation, we included a supervised baseline and reported the standard error of the mean (SEM) for all metrics to strengthen the statistical grounding of our results.
> > >
> > > The supervised baseline achieves an SI-SNR of 18.21 ± 0.27 dB, which is significantly higher than our best unsupervised performance (15.23 ± 0.26 dB for SURF Self-Remixing). This gap of roughly 3 dB clearly demonstrates that the task is not yet saturated and that substantial room for improvement remains.
> > >
> > > Furthermore, the performance gains achieved by SURF (Self-Remixing) over the MixIT baseline (14.18 ± 0.20 dB) are statistically significant. With a sample size of N=1000, the standard error is sufficiently small to confirm that SURF’s improvements represent a robust and meaningful step toward the supervised upper bound.

---

### Decision · Program_Chairs · 2026-04-30

**Decision:**

Accept (regular)

**Comment:**

The paper initially received mixed ratings, but the reviewers were convinced by the rebuttal and two reviewers have increased their score. There is now a consensus to accept the paper. Based on the reviews and discussions, the area chair agrees with the reviewers' assessment and follows their recommendation.